Report

# Oncogenic YAP sensitizes cells to CHK1 inhibition via CDK4/6 driven G1 acceleration

Dörthe Gertzmann [1], Cornelius Presek[1], Anna Lena Mattes[1], Marco Sänger [1], Marie Zoller[1], Christina Schülein-Völk[2], Carsten P Ade [3], Martin Eilers [3] & Stefan Gaubatz [1✉]

## Abstract

Replication stress is a driver of genomic instability, contributing to carcinogenesis by causing DNA damage and mutations. While YAP, the downstream co-activator of the Hippo signaling pathway, plays a crucial role in regulating cell growth and differentiation, it is unclear whether it generates replication stress exploitable for therapy. Here, we report that oncogenic YAP shortens the G1 phase through increased CDK4/6 activity, leading to early S-phase entry. This causes origin underlicensing, an overall reduced rate of DNA replication, and, unusually, an accelerated speed of individual replication forks. CHK1 inhibition in cells expressing oncogenic YAP results in DNA damage during S-phase, which is not due to premature CDK1 activation or mitotic entry. Sensitivity to CHK1 inhibition depends on the YAP-TEAD interaction and involves a global increase in transcription and an increase in transcription–replication conflicts (TRCs). Replication stress from oncogenic YAP can be mitigated by restoring G1 length through partial CDK4/6 inhibition or by reducing YAP-induced hypertranscription. Our findings suggest a potential therapeutic strategy for targeting YAP-dependent cancers by exploiting their vulnerability to replication stress.

**Keywords** YAP; Replication Stress; CHK1; CDK4/6
**Subject Categories** Cancer; Cell Cycle; DNA Replication, Recombination & Repair

## Introduction

Yes-associated protein (YAP) and the related TAZ protein are transcriptional coactivators that act as key effectors of the Hippo pathway (Lopez-Hernandez et al, 2021; Totaro et al, 2018). The Hippo signaling cascade consists of the core kinases MST1/2 and LATS1/2, and their regulatory subunits Salvador (SAV1) and MOB1 (Moya and Halder, 2019). When the Hippo pathway is

active, LATS kinases phosphorylate YAP and TAZ, leading to their cytoplasmic retention and subsequent degradation via the proteasome. In contrast, when Hippo signaling is suppressed, unphosphorylated YAP translocates into the nucleus, where it associates with TEAD transcription factors to drive the expression of genes involved in proliferation, survival, and stemness. Mutations in Hippo pathway components, such as amplification of YAP, or loss of the tumor suppressors NF2, LATS1 or LATS2, are frequently found in human tumors (Calses et al, 2019; Franklin et al, 2023; Zanconato et al, 2016). Genetic alterations such as YAP amplification or loss of the tumor suppressors NF2, LATS1, or LATS2 result in YAP/TAZ hyperactivation, promoting uncontrolled cell growth and malignant transformation. Aberrant YAP/TAZ activation has been implicated in tumor initiation and progression as well as in therapy resistance and is generally correlated with a poor outcome (Thompson, 2020). Given its oncogenic role, targeting YAP-TEAD is a promising therapeutic strategy for cancer treatment (Dey et al, 2020). Several small molecules and peptides that disrupt this interaction are under investigation, some showing promising results in preclinical models (Franklin et al, 2023; Hagenbeek et al, 2023; Pobbati et al, 2023).

High rates of cell proliferation by overexpression or activation of oncogenes can lead to replication stress, a condition characterized by a slowdown or stalling of the replication fork (Saxena and Zou, 2022). Cancer cells cope with DNA damage and replication stress by upregulating the replication checkpoint response and activation of the kinase ATR and the downstream effector kinase CHK1 (Cimprich and Cortez, 2008). CHK1 phosphorylates and promotes the degradation of CDC25A, thereby reducing CDK2 activity, thus restricting origin firing and pausing S-phase to allow for DNA repair and resolution of replication conflicts (Zhang and Hunter, 2014). In addition, CHK1 also inhibits CDC25C and activates WEE1, thereby inhibiting CDK1 activity and preventing an unscheduled G2/M transition (Sorensen and Syljuasen, 2012). When replication stress is not detected by the checkpoint and cell cycle progression continues, cells can undergo excessive DNA damage and die (Beck et al, 2010; Buisson et al, 2015; Toledo et al, 2013). On the other hand, underreplicated DNA and unresolved replication stress can lead to aberrant mitosis and chromosome instability and can contribute to tumor progression (Gaillard et al, 2015).

[1]Department of Biochemistry and Cell Biology, Theodor Boveri Institute, Biocenter, Julius Maximilian University Würzburg, Am Hubland, 97074 Würzburg, Germany. [2]Core Unit High-Content Microscopy, Theodor Boveri Institute, Biocenter, Julius Maximilian University Würzburg, Am Hubland, 97074 Würzburg, Germany. [3]Department of Biochemistry and Molecular Biology, Theodor Boveri Institute, Biocenter, Julius Maximilian University Würzburg, Am Hubland, 97074 Würzburg, Germany.
✉E-mail: stefan.gaubatz@uni-wuerzburg.de

Oncogenes such as MYC and Cyclin E promote replication stress by inducing excessive replication initiation, leading to the activation of more origins of replication than normal (Bartkova et al, 2005; Bartkova et al, 2006; Di Micco et al, 2006; Macheret and Halazonetis, 2018). This increased origin firing depletes the available pool of replication factors and nucleotides, causing a shortage and stalling of the replication process (Gaillard et al, 2015). Overexpression of oncogenic Cyclin E can also promote inappropriate S-phase entry, leading to incomplete DNA replication origin licensing and underreplication (Ekholm-Reed et al, 2004). A third mechanism of oncogene-induced replication stress is an increase in collisions between replication forks and the transcription machinery, or transcription–replication conflicts, leading to secondary nucleotide structure such as R-loops. This can result in replication fork stalling and collapse (Gomez-Gonzalez and Aguilera, 2019; Jones et al, 2013). While MYC, RAS, and Cyclin E have been well-documented in inducing replication stress (Forment and O'Connor, 2018; Matthews et al, 2022), the role of YAP in this context remains largely unexplored. Recent work in Xenopus egg extracts has shown that YAP can bind to and cooperate with RIF1 to regulate the replication timing program, implicating YAP as a potential modulator of origin firing and DNA replication dynamics (Melendez Garcia et al, 2022). These findings suggest that YAP may contribute to replication stress through mechanisms distinct from classical oncogenes.

Here we show that YAP promotes proliferation and accelerates G1 progression leading to underlicensing of replication origins resulting in an overall reduction in DNA synthesis but increased replication fork speed. While expression of oncogenic YAP in untransformed cells does not cause DNA damage on its own, it increases sensitivity to ATR and CHK1 inhibition. This sensitizing effect of oncogenic YAP is not due to premature mitotic entry but results from an unusual type of replication stress with fewer $Mcm_{2-7}$ complexes loaded on DNA and fewer replication origins activated, but replication forks proceeding at enhanced speeds. In addition, YAP increases collisions between the transcription machinery and DNA replication. Replication stress and the increased sensitivity to ATR/CHK1 inhibition could represent a vulnerability of YAP-dependent cancers that can be exploited therapeutically.

# Results and discussion

## Oncogenic YAP promotes proliferation and shortens G1

To investigate how oncogenic YAP alters cell cycle progression and replication dynamics, we used untransformed epithelial MCF10A cells expressing doxycycline-inducible YAP5SA, a constitutive active allele of YAP that cannot be inhibited by the Hippo kinases (Zhao et al, 2010; Zhao et al, 2007). We and others have previously shown that YAP5SA promotes oncogenic transformation of MCF10A cells (Pattschull et al, 2019; Zanconato et al, 2015). Without doxycycline induction, MCF10A cells express very low levels of endogenous YAP, which is predominantly cytoplasmic (Fig. 1A). YAP5SA was robustly induced by doxycycline treatment for 48 h and localized to the nucleus. Proliferation of YAP5SA-overexpressing cells was accelerated (Fig. 1B). Compared to wild-type MCF10A and uninduced MCF10A-YAP5SA cells, YAP5SA-

expressing cells exhibited more than double the relative growth over 10 days of culture. To identify the cause of increased proliferation of YAP5SA-expressing cells, we determined the percentage of cells in each cell cycle phase. YAP5SA expression for 48 h resulted in a significant decrease in the proportion of cells in G1, while the timing of the S to M progression remained unchanged, indicating that YAP shortens the G1 phase progression (Fig. 1C,D). Notably, despite premature S-phase entry, YAP5SA-expressing cells exhibited accelerated overall proliferation compared to control cells (Fig. 1B). In contrast, overexpression of Cyclin E1, which also shortens G1 (Fig. 1E), resulted in a slowdown of proliferation already one day after induction (Fig. 1F).

## Oncogenic YAP shortens G1 by activating CDK4/CDK6

To understand how oncogenic YAP shortens G1, we examined the molecular mechanism underlying G1 progression. In G1, Cyclin D–CDK4/6 complexes initially phosphorylate RB, followed by further phosphorylation by Cyclin E–CDK2, releasing E2F transcription factors and promoting cell cycle progression (Rubin et al, 2020). To determine CDK2 and CDK4/6 activity, we used previously established reporter constructs that are specifically responsive to these kinases (Spencer et al, 2013; Yang et al, 2020) (Figs. 2A and EV1A). CDK2- or CDK4/6-dependent phosphorylation induces translocation of the respective reporter from the nucleus to the cytoplasm. Accordingly, CDK2 and CDK4/6 activity can be quantified at the single-cell level by measuring the relative localization of the reporters between the cytoplasm and nucleus (Fig. 2A). Comparison of control cells with YAP5SA-expressing cells revealed an elevation of CDK4/6 activity during G1, while there was a weaker effect in S and G2 (Figs. 2B and EV1B). CDK2 activity remained unchanged in G1, S and G2 (Figs. 2B and EV1B). This indicates that YAP's effect is more specific to CDK4/6 regulation during G1, rather than a global alteration of CDK activity throughout the entire cell cycle. Partial inhibition of CDK4/6 with low doses of palbociclib restored G1 length of YAP5SA-expressing cells, confirming that enhanced CDK4/6 activity is crucial for the accelerated S-phase entry induced by YAP (Fig. 2C). Reporter activity measurements across a range of palbociclib concentrations confirmed that the drug selectively inhibits CDK4/6, with only minimal impact on CDK2 activity (Figs. 2D and EV1C). Importantly, these measurements also indicated that CDK4/6 activity remains elevated in YAP5SA-expressing cells even after palbociclib treatment. The increase in CDK4/6 activity in G1 by YAP5SA was paralleled by an increase in the ratio of phosphorylated RB to total RB (Fig. 2E). Palbociclib reduced RB phosphorylation more strongly in control cells than in YAP5SA-expressing cells. In S-phase, the effect of palbociclib on RB phosphorylation was weaker, as phosphorylation is maintained after cells pass the restriction point (Fig. EV1D). Density plots confirmed that palbociclib induced a stronger leftward shift in control cells than in YAP5SA-expressing cells in G1 (Fig. EV1E). Taken together, these analyses confirm that YAP sustains CDK4/6-dependent RB phosphorylation in G1, even in the presence of low concentrations of palbociclib. In addition, western blotting confirmed that phosphorylation of RB was blocked by palbociclib and increased when YAP5SA was expressed for 24 h prior to addition of the CDK4/6 inhibitor for 24 h (Fig. 2F). Consistent with these findings, the inhibitory effect of palbociclib on the expression of E2F-target

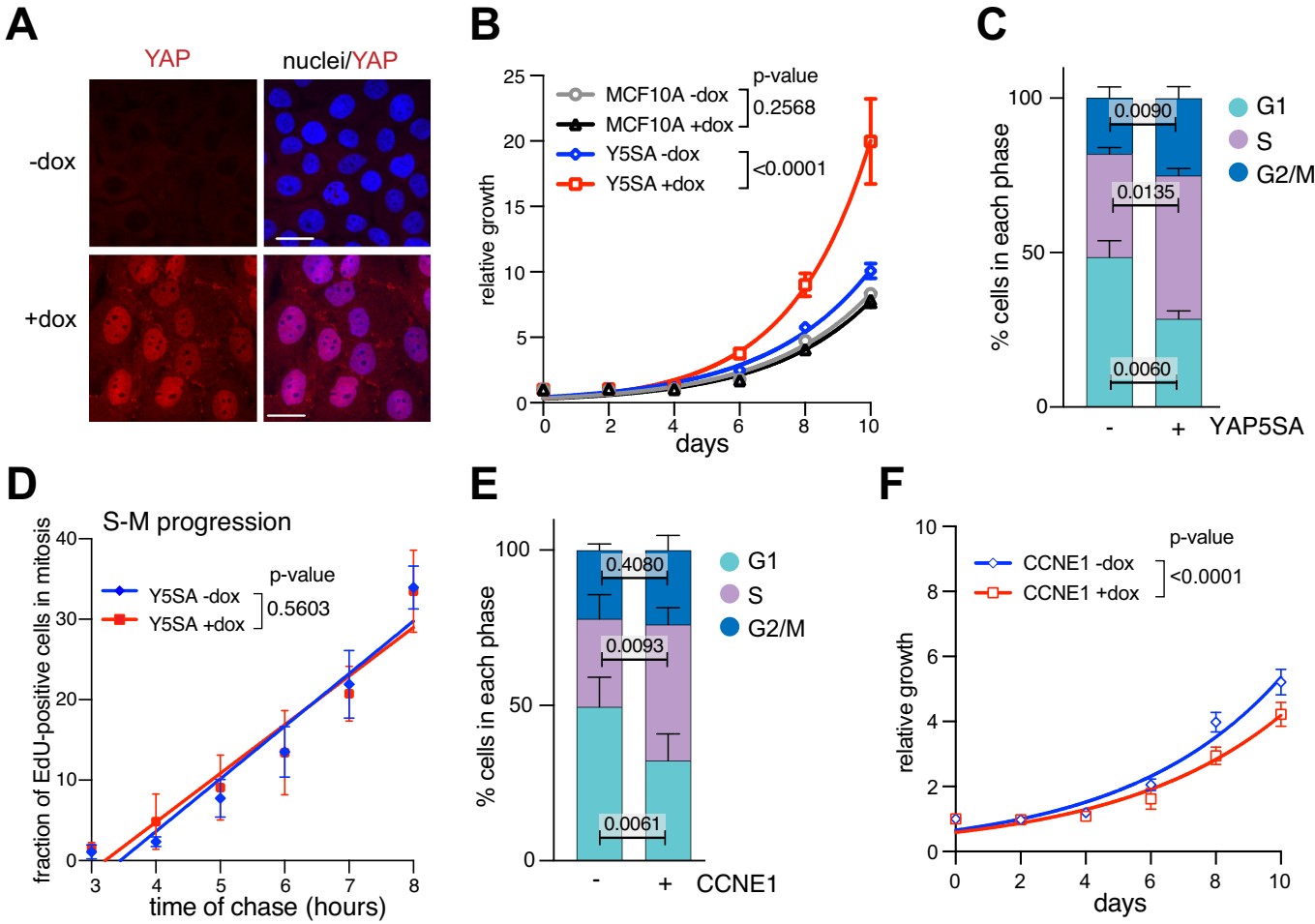

**Figure 1. Oncogenic YAP promotes proliferation and shortens G1.**

(A) Immunostaining of YAP in MCF10A cells expressing doxycycline-inducible YAP5SA. dox doxycycline. (B) Growth curve of parental MCF10A and MCF10A-YAP5SA cells treated with doxycycline as indicated. Mean $+/-$ SEM. $P$ values were calculated using the extra sum-of-squares F test ($n = 3$ independent replicates). (C) Cell cycle distribution of cells 48 h after YAP5SA induction was analyzed by high-content microscopy. Mean $+/-$ SD. Students $t$ test ($n = 3$ independent replicates). (D) A EdU pulse-chase experiment was performed to measure the S to M transition. Cells were pulse-treated with 10 μM EdU for 20 min and chased for the indicated times. Cells were then stained for the mitosis marker pH3 (S10). The fraction of EdU-positive entering mitosis (pH3-positive) was plotted against the time of the chase. Mean $+/-$ SD. $P$ values were calculated using the extra sum-of-squares F test ($n = 3$ independent replicates). (E) Cell cycle distribution of CCNE1-expressing cells treated for 48 h with doxycycline. Mean $+/-$ SD. Students $t$ test ($n = 3$ independent replicates). (F) Growth curve of MCF10A cells expressing doxycycline-inducible CCNE1. Mean $+/-$ SEM. $P$ values were calculated using the extra sum-of-squares F test ($n = 3$ independent replicates). Source data are available online for this figure.

genes was attenuated in YAP5SA-expressing cells (Fig. 2G). YAP5SA expression resulted in a modest upregulation of both cyclin D and CDK6 at the mRNA and protein level (Fig. EV1F,G), suggesting that YAP5SA may directly or indirectly enhance the expression of these components, potentially driving increased CDK4/6 activity. Given the relatively modest induction observed, it is likely that additional, indirect mechanisms also contribute to the overall increase in CDK4/6 activity.

To test whether YAP is also sufficient to overcome an established G1 arrest, we first inhibited CDK4/6 with 2 μM palbociclib for 48 h before induction of YAP5SA. As expected, palbociclib arrested cells in G1. Induction of YAP5SA during the final 24 h was sufficient to increase the fraction of cells in S-phase and reduce the G1 fraction (Fig. 2H). Western blot analysis showed that under these conditions, RB phosphorylation was inhibited by palbociclib but partially restored by YAP5SA when

it was expressed after CDK4/6 inhibition (Fig. 2I). YAP5SA increased the expression of E2F-target genes in palbociclib-treated cells (Fig. 2J). YAP also reactivated the expression of E2F-target genes in response to Nutlin-3 or doxorubicin, two additional drugs known to promote cell cycle arrest and repress cell cycle genes via E2F-RB and DREAM complexes (Fig. 2K,L). These results suggest that YAP acts on the CDK4/6 axis to promote the G1/S transition.

## YAP5SA leads to underlicensing and accelerated DNA replication

We next investigated whether shortened G1 and accelerated S-phase entry by YAP5SA affects DNA replication dynamics. We first assessed replication origin licensing, which occurs in G1 and could be affected by premature S-phase entry. To do so we

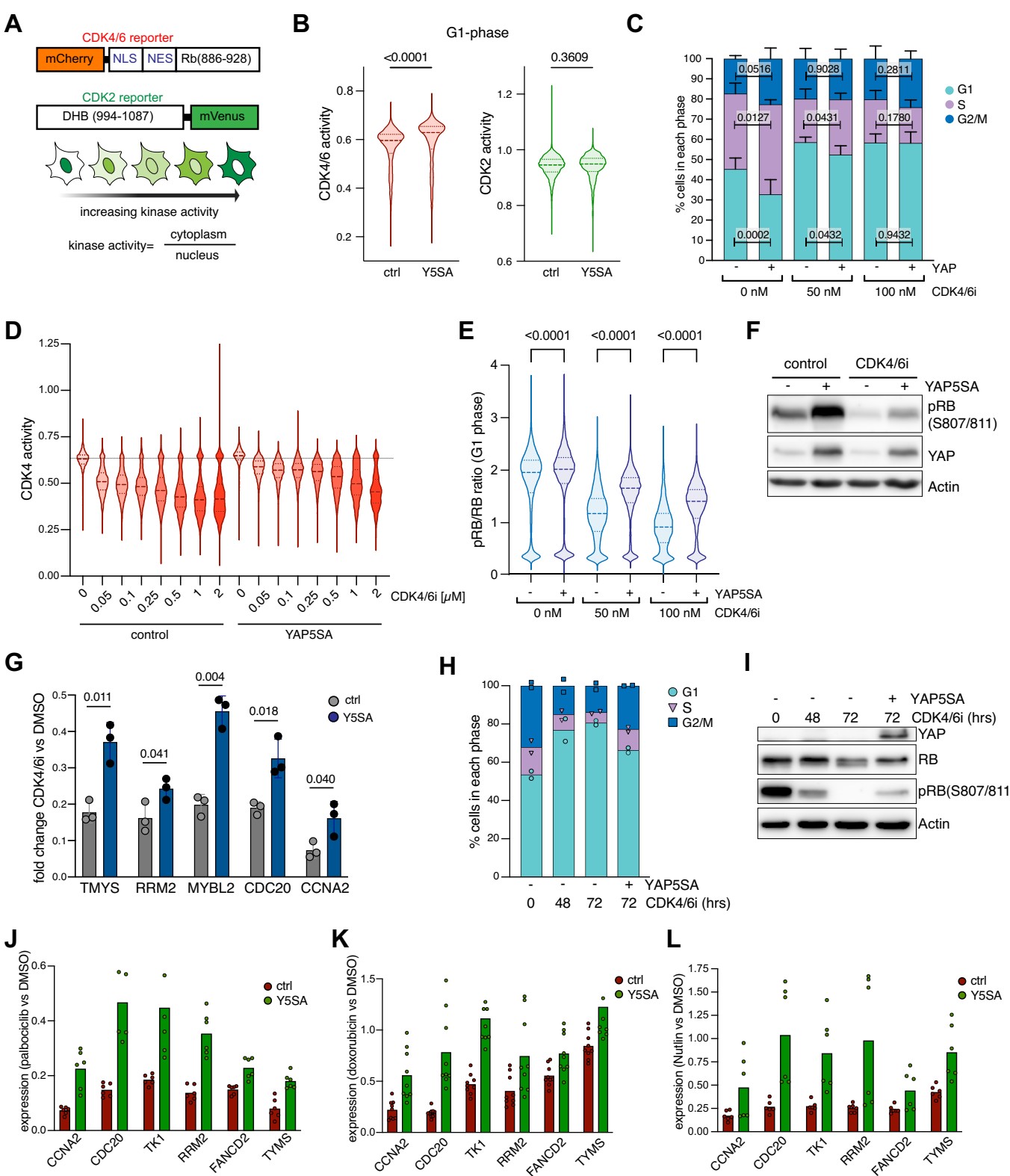

◄ **Figure 2. Oncogenic YAP shortens G1 by CDK4/CDK6 activation.**

(A) Scheme of CDK4/6 and CDK2 reporter constructs. Increased CDK kinase activity promotes an increase in the relative cytoplasmic versus nuclear localization of the CDK-reporter. (B) CDK4/6 and CDK2 in G1 phase of control and YAP5SA-expressing MCF10A cells. Violin plots display single-cell measurements with ≥2659 cells analyzed per condition. $P$ values were calculated using unpaired Student's $t$ test ($n = 3$ independent replicates). (C) MCF10A control cells and YAP5SA-expressing cells were treated with the indicated concentrations of palbociclib for 24 h. The fraction of cells in each phase of the cell cycle was determined by EdU labeling and high-content microscopy. Mean $+/−$ SD. $P$ values were calculated using ordinary one-way ANOVA ($n = 5$ independent replicates). (D) CDK4/6 activity of control and YAP5SA-expressing MCF10A cells treated with increasing concentrations of palbociclib. Violin plots display single-cell measurements with ≥1038 cells analyzed per condition ($n = 3$ independent replicates). (E) Violin plots showing the ratio of phospho-RB to total RB in S-phase cells treated with the indicated concentrations of palbociclib. S-phase cells were identified by EdU incorporation. Single-cell measurements from a representative experiment with at least 3543 cells analyzed per condition. $P$ values were calculated using ordinary one-way ANOVA ($n = 3$ independent replicates). (F) Immunoblot of phospho-RB (S807/811) in control cells and YAP5SA-expressing cells treated with palbociclib as indicated. (G) RT-qPCR was used to determine fold changes in mRNA levels of the indicated E2F-target genes in cells treated with 0.5 µM palbociclib for 24 h. Mean $+/−$ SD. $P$ values were calculated using an unpaired Student's $t$ test ($n = 3$ independent replicates). (H) FACS analysis of cells treated with palbociclib for 72 h. During the final 24 h, expression of YAP5SA was induced by doxycycline as indicated. Samples were collected after 48 h of palbociclib treatment to verify that cells were fully arrested at the time of YAP5SA induction ($n = 2$ independent replicates). (I) Immunoblot of samples as treated as in (H) demonstrating that YAP5SA restores RB phosphorylation in cells treated with palbociclib. (J) RT-qPCR was used to determine fold changes in mRNA levels of the indicated RB and DREAM-regulated E2F-target genes upon palbociclib treatment and YAP5SA induction. The data set contains two biological replicates, and each one was analyzed with three technical replicates. (K, L) As in (J), but cells were treated with doxorubicin (K) or Nutlin-3 (L) for 24 h. The data set contains two to three biological replicates, and each one was analyzed with three technical replicates. Source data are available online for this figure.

measured the loading of minichromosome maintenance (MCM) complexes onto DNA. We pulse-labeled replicating cells with EdU, stained for DNA content using Hoechst and for chromatin-bound MCM7, a subunit of the MCM$_{2-7}$ complex (Fig. EV2A). High-content microscopy was then used to quantify chromatin-bound MCM7 at the single-cell level in early S-phase as a measure of licensing, since MCM loading occurs in G1 and is unloaded in S-phase when replication terminates (Mendez and Stillman, 2000). Expression of YAP5SA for 48 h significantly reduced loaded, chromatin-bound MCM7, indicating that YAP leads to under-licensing (Fig. 3A,B). This reduction in MCM7 loading was not due to changes in the expression levels of MCM7 (Fig. EV2B). A similar reduction in chromatin-bound MCM7 was also observed in mid-S-phase, consistent with a reduction in origins available for activation (Fig. EV2C). In late S-phase and G2, chromatin-bound MCM7 levels were lower overall, and the YAP-induced effect was smaller, as expected due to the normal unloading of MCM during replication completion (Fig. EV2C).

We next asked whether reduced licensing affects DNA replication dynamics. Plotting EdU intensities revealed that YAP-expressing cells exhibit a significant decrease in total EdU intensities per cell compared with control cells, suggesting an overall reduction in DNA synthesis rate (Fig. 3C). The reduction in EdU incorporation was rescued by inhibition of CHK1 with prexasertib, which is likely due to the excess origin firing triggered by the loss of CHK1 activity (Maya-Mendoza et al, 2007; Petermann and Caldecott, 2006; Syljuåsen et al, 2005). YAP5SA-expressing cells remained underlicensed even after prexasertib treatment, demonstrating that CHK1 inhibition does not correct the licensing defect in these cells (Fig. 3A,B). Thus, reduced licensing due to accelerated S-phase entry likely explains the decreased overall EdU incorporation rate in YAP-expressing cells, as the number of active replication origins during DNA replication is insufficient under these conditions. CHK1 phosphorylated at serine 296, a marker of its kinase activity, was modestly increased in YAP5SA-expressing cells (Fig. EV2D). This elevated CHK1 activity could limit origin firing, thereby contributing to dampenend DNA synthesis.

Cells expressing Cyclin E also exhibited underlicensing and reduced EdU intensities (Fig. 3D). However, CHK1 inhibition failed to restore EdU incorporation when Cyclin E was over-produced (Fig. 3E). A likely explanation for this difference is that Cyclin E overexpression depletes the nucleotide pool (Bester et al, 2011; Jones et al, 2013), whereas YAP has been shown to maintain dNTP levels by upregulating key enzymes involved in nucleotide metabolism, such as RRM2 (Santinon et al, 2018). This suggests that nucleotide levels are sufficiently high in YAP-expressing cells to support DNA synthesis even under CHK1 inhibition.

To directly investigate DNA replication, we measured replication fork speed using DNA fiber assays by labeling replication forks through sequential incorporation of two nucleotide analogs. Upon spreading of the DNA on glass slides, the tracks were visualized using antibodies specific to the labels. This enables determination of the length of replication tracts and quantification of replication fork progression (Fig. 3F). Surprisingly, YAP expression was associated not with a decrease, but with a small yet reproducible and significant increase in fork speed (Fig. 3G). CHK1 inhibition resulted in strong inhibition of replication fork speed, as expected and previously reported (Petermann and Caldecott, 2006), which was also partially overcome by YAP. Replication stress can also lead to asymmetric sister forks progression emanating from the same origin. However, we found no significant increase in fork asymmetry after YAP expression (Fig. 3H). In contrast, CHK1i resulted in a slight increase in asymmetric fork movement, consistent with previous reports (Besteiro et al, 2019). While replication stress is typically associated with slower fork speed, it has also been associated with increased fork speed (Maya-Mendoza et al, 2018). Notably, elevated fork speed itself can contribute to replication stress and genomic instability. Since origin activity and fork speed are interdependent (Rodriguez-Acebes et al, 2018; Zhong et al, 2013), the reduced availability of replication origins in YAP5SA-overexpressing cells may lead to a compensatory increase in fork speed—similar to what has been observed upon depletion of MCMBP or the Cyclin D regulator AMBRA1 (Maiani et al, 2021; Sedlackova et al, 2020). Furthermore, YAP-induced acceleration of fork speed may also be related to E2F activation (Ehmer and Sage, 2016; Kapoor et al, 2014), as E2F-dependent transcription has been shown to regulate overall replication capacity and the rate of DNA synthesis (Pennycook et al, 2020).

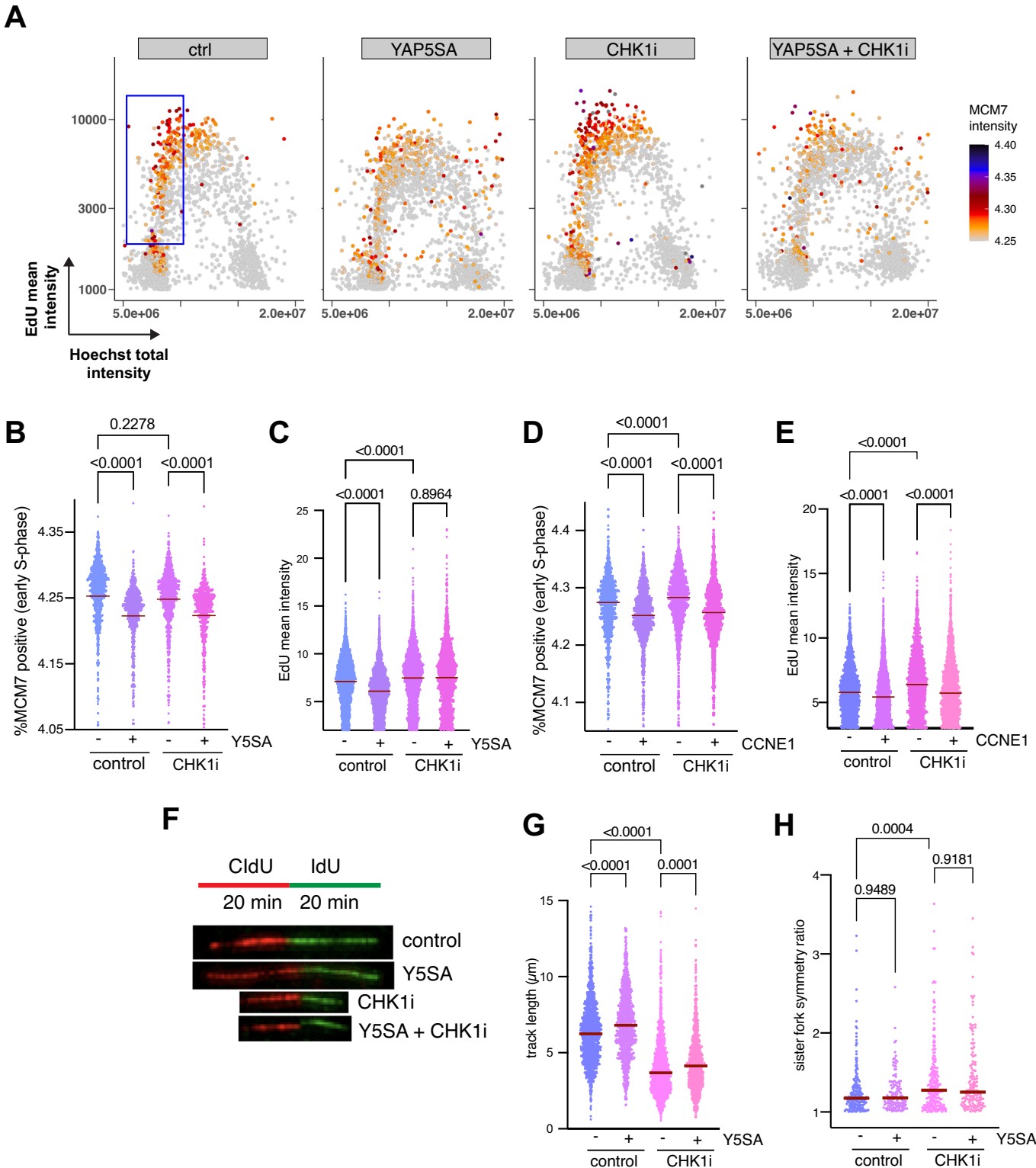

**Inhibition of CHK1 in YAP-expressing cells leads to DNA damage during S-phase**

We next asked whether altered replication dynamics by YAP result in DNA damage. To address this question, we investigated

phosphorylation of KAP1 at S824, a marker of ATM-mediated response to double-strand breaks. Phosphorylation of KAP1 was not detected by expression of YAP5SA alone (Fig. 4A). Because cells that experience replication stress can be particularly sensitive to ATR and CHK1 inhibitors, we investigated DNA damage upon

**Figure 3. YAP5SA leads to underlicensing and to changes in DNA replication.**

(A) High-content microscopy-based analysis of chromatin-bound MCM7. S-phase cells were labeled with EdU. We assessed for DNA content (*x* axis), EdU incorporation (*y* axis) and MCM7. Each dot represents a single cell and is color-coded according to mean MCM7 fluorescent intensity. For each sample, 3000 cells were randomly selected from a representative experiment (*n* = 3 independent replicates). (B) Chromatin-bound MCM7 in early S-phase. Violin plots display single-cell measurements from a representative experiment, with ≥748 cells analyzed per condition (*n* = 3 independent replicates). (C) EdU intensities of YAP5SA-expressing cells. Violin plots display single-cell measurements from a representative experiment, with ≥2196 cells analyzed per condition (*n* = 3 independent replicates). (D) Chromatin-bound MCM7-positive cells in early S-phase after expression of CCNE1. Violin plots display single-cell measurements from a representative experiment, with ≥1466 cells analyzed per condition (*n* = 3 independent replicates). (E) EdU intensities of CCNE1-expressing cells. Single-cell measurements from a representative experiment, with ≥2358 cells analyzed per condition (*n* = 3 independent replicates). (F–H) DNA fiber assays of cells sequentially labeled with IdU and CldU. (F) Labeling scheme and representative DNA fiber images. (G) Quantification ((IdU+CldU)/2) track lengths) of unidirectional DNA replication structures of at least 145 fibers per condition (*n* = 3 independent replicates). (H) Sister fork asymmetry (longer/shorter sister fork length). At least 39 bidirectional replication structures were measured per condition per experiment (*n* = 3 independent replicates). (B–E, G, H) *P* values were calculated using ordinary one-way ANOVA. Source data are available online for this figure.

treatment with the CHK1 inhibitor prexasertib. Indeed, robust KAP1 phosphorylation was detected following CHK1 inhibition specifically in YAP5SA-expressing cells (Fig. 4A). Notably, pRPA32 (Ser33), which is phosphorylated in response to replication stress, was also detected in YAP-expressing cells treated with CHK1i but was absent in control cells.

To investigate whether DNA damage by YAP restricted to specific cell cycle phases, we performed high-content microscopy at the single-cell level. YAP5SA alone did not induce detectable DNA damage beyond background levels in any cell cycle phase. Following a 2-h treatment with a CHK1 inhibitor, we observed an increase in cells positive for the DNA-damage marker γH2A.X in S-phase (Figs. EV3A and 4B). Notably, YAP5SA induction significantly increased the CHK1i-induced γH2A.X-positive population in S-phase, suggesting that YAP enhances sensitivity to CHK1 inhibition during active DNA replication (Fig. 4B,C).

One key function of the ATR-CHK1 checkpoint is to inhibit the S/G2 transition, thereby delaying mitotic entry (Saldivar et al, 2018). Mitotic gene expression is mainly driven by FOXM1 and by the B-MYB-MuvB (MMB) complex (Muller et al, 2022), which are necessary for premature mitosis and for triggering replication catastrophe following CHK1 inhibition in certain cancer cell lines (Blosser et al, 2020; Branigan et al, 2021; Chung et al, 2019). Because our previous research demonstrated that YAP cooperates with the MMB-complex to activate G2/M cell cycle genes (Jessen et al, 2024; Pattschull et al, 2019), we asked whether unscheduled activation of the G2/M transcriptional program contributes to increased sensitivity to CHK1 inhibition. In support of this idea, we observed premature accumulation of Cyclin B by YAP5SA during S-phase (Figs. 4D and EV3B). However, despite upregulation of Cyclin B and other mitotic genes by YAP5SA (Pattschull et al, 2019), this was insufficient to trigger premature CDK1 activation following CHK1 inhibition, as evidenced by the absence of phospho-Lamin A/C staining at S22, a known CDK1 substrate (Figs. 4E and EV3B). Furthermore, staining for Histone H3 phosphorylated on Ser10 showed that YAP5SA did not lead to premature mitosis (Figs. 4F and EV3B). In contrast, WEE1 depletion, used as a control, but not PKYMT1 depletion, resulted in robust premature CDK1 activation in EdU-positive cells (Fig. EV3C–E).

We conclude that YAP sensitizes cells to CHK1 inhibition independently from an unscheduled mitosis during S-phase. Consistent with this notion, prolonged CHK1 inhibition (4 h and 8 h) led to severe replication defects with low levels of EdU

incorporation but with high levels of the DNA-damage markers γH2AX and pKAP1 (Fig. 4G).

## Oncogenic YAP sensitizes cells to CHK1 and ATR inhibition

Given that YAP5SA increases the DNA damage upon CHK1 inhibition, we next asked whether YAP affects cell viability following ATR-CHK1 pathway inhibition. We first assessed cell viability following exposure to increasing doses of the CHK1 inhibitor prexasertib. YAP5SA-expressing cells exhibited significantly reduced viability compared to control MCF10A cells upon CHK1 inhibitor treatment (Fig. 5A). The response of YAP5SA-expressing cells to prexasertib was comparable to the sensitizing effect observed with Cyclin E1 expression (Fig. 5B). YAP5SA also enhanced sensitivity to the ATR inhibitor ceralasertib (AZD6738) (Fig. 5C). However, YAP5SA did not alter sensitivity to WEE1 inhibition with advosertib (AZD1775), which primarily regulates the G2/M transition by directly activating CDK1 and CDK2 (Fig. 5D).

YAP acts as a co-activator that predominantly associates with enhancers through its interaction with the TEAD family of transcription factors (Piccolo et al, 2014; Totaro et al, 2018). To test whether the synergy between CHK1 inhibition and YAP depends on the interaction between YAP and TEAD transcription factors, we generated a mutant of YAP, S94A, which cannot bind to TEAD and with impaired transcriptional activation (Zhao et al, 2008) (Fig. EV4A). MCF10A cells expressing the compound mutant of YAP5SA and S94A were not sensitized to CHK1 inhibition (Fig. 5E). Moreover, inhibition of CHK1 in these cells did not increase levels of phosphorylated KAP1 (Fig. 5F). These data indicate that increased vulnerability to CHK1 inhibition by YAP depends on YAP-TEAD transcriptional activity.

To more directly test the role of transcription in YAP-induced DNA damage, we labeled nascent RNA with ethynyl uridine (EU). Because changes in cell cycle distribution may affect transcription rates, we wanted to quantify nascent transcription specifically in S-phase and early G2. To do so, we employed PIP-FUCCI, an improved version of the original FUCCI (Fluorescent Ubiquitination-based Cell Cycle Indicator) system (Fig. EV4B) (Grant et al, 2018). PIP-FUCCI uses a PIP-degron that is rapidly degraded upon the beginning of DNA replication and gradually regains fluorescence in G2/M. We pulse-labeled PIP-FUCCI cells with ethynyl uridine (EU), fixed the cells, labeled EU by click chemistry, performed high-content microscopy and gated cells for

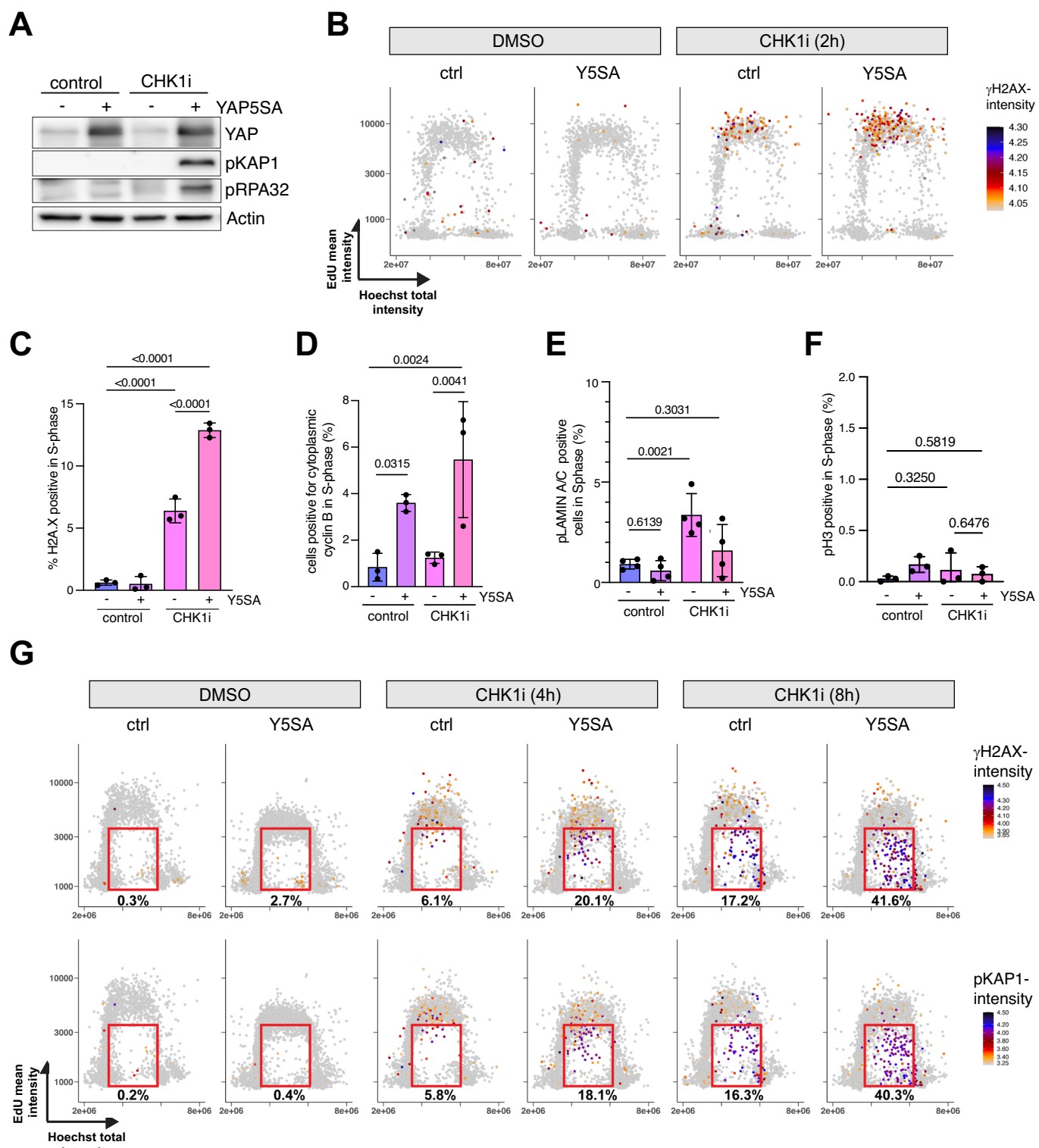

S-phase and early G2. These experiments revealed a global increase in the transcription rate by YAP5SA during these phases (Fig. 5G). Treatment with 5,6-dichloro-1-b-ᴅ-ribofuranosylbenzimidazole (DRB), an inhibitor of transcriptional elongation, reduced the EU signal of YAP-expressing cells (Fig. 5G). More importantly, DRB also reduced γH2AX staining, indicating that DNA damage in these cells depends on ongoing transcription (Fig. 5H).

The increase in replication speed and hypertranscription by YAP5SA may lead to transcription–replication conflicts (TRCs). Such conflicts occur when the processes of transcription and DNA

**Figure 4. Inhibition of CHK1 in YAP5SA-expressing cells results in DNA damage in S-phase.**

(A) Immunoblot of the DNA-damage marker pKAP1(S824) and the replication stress marker pRPA32(S33). Cells were treated with 100 nM CHK1i for 8 h as indicated. (B) High-content microscopy-based analysis of cells treated for 2 h with 100 nM CHK1i as indicated. S-phase cells were labeled with EdU. We assessed for DNA content (x axis), EdU incorporation (y axis) and γH2AX. Each dot represents a single cell and is color-coded according to mean γH2AX fluorescent intensity. For each sample, 2900 cells were randomly selected from a representative experiment (n = 3 independent replicates). (C) Quantification of γH2AX in the experiment shown in (B). Mean +/− SD. P values were calculated using ordinary one-way ANOVA (n = 3 independent replicates). (D–F) Percentage of cells positive for cytoplasmic Cyclin B, pLAMIN A/C (S22) or pH3 (S10) in S-phase. Mean +/− SD. P values were calculated using ordinary one-way ANOVA (n = 3 independent replicates). See also Figure EV3B. (G) Cells were treated for 4 h and 8 h with CHK1i and stained for the DNA-damage markers γH2AX (upper panels) and pKAP1 (lower panels). The percentage of γH2AX- and pKAP1-positive cells in intra-S-phase with low EdU incorporation is indicated. For each sample, 4000 cells were randomly selected from a representative experiment (n = 3 independent replicates). Source data are available online for this figure.

replication collide or impede each other, posing an obstacle to replication progression (Bowry et al, 2021; Lin and Pasero, 2021; Ngoi et al, 2021; Primo and Teixeira, 2019). To test for TRCs, we performed proximity ligation assays (PLA) between PCNA, the sliding clamp of DNA replication, and the elongating RNA Pol II phosphorylated at Serine 2. We found that induction of YAP5SA for 48 h indeed increased the proximity between transcription and replication machineries (Figs. 5I and EV4C). CHK1 inhibition further elevated the contacts between PCNA and RNA Pol II in YAP5SA-expressing cells, consistent with the observed increase in DNA damage under these conditions.

These findings are consistent with prior work in a developmental context, where deletion of the Hippo kinases LATS1 and LATS2 during mouse brain development was associated with an increase in global transcription, replication stress, and DNA damage (Lavado et al, 2018). This supports the idea that deregulated YAP activity can drive excessive transcription during S-phase, contributing to replication–transcription conflicts and DNA damage.

Hypertranscription and unresolved TRCs can result in R-loop formation, which are structures that are formed when an RNA molecule hybridizes with its DNA template strand, leaving a displaced single-stranded DNA loop (Bowry et al, 2021; Garcia-Muse and Aguilera, 2019). R-loops can cause single and double DNA strand breaks and contribute to replication stress. Slot blot assays using an antibody that recognizes RNA/DNA-hybrids (S9.6) revealed reduced levels of R-loop formation upon expression of YAP (Fig. EV4D,E). The S9.6 signal disappeared when the extracted genomic DNA was digested with RNase H1, confirming the specificity of the assay. Because S9.6-based detection of R-loops has been discussed controversially (Chédin et al, 2021), we also performed MapR, an antibody-independent method that uses a catalytically inactive RNase H1 fused to micrococcal nuclease to directly map R-loop structures genome-wide (Yan et al, 2019). MapR assays confirmed that R-loop formation is not increased upon expression by YAP5SA either alone or in combination with CHK1 inhibition (Fig. EV4F). In contrast, inhibition of BRD4 by treatment with JQ1 resulted in an increase in R-loops, consistent with previous reports and confirming the specificity of the assay (Edwards et al, 2020; Lam et al, 2020). Thus, despite increased global transcription and an increase in TRCs, YAP5SA expression did not lead to R-loop accumulation.

Finally, we sought to determine whether accelerated S-phase entry, which disrupts the separation of transcriptionally active G1 and replication-dominant S-phase, contributes to the DNA-damage phenotype observed with YAP5SA-expressing cells. To test the idea that a shortened G1 phase underlies increased CHK1i sensitivity by

oncogenic YAP, we artificially prolonged the G1 phase by inhibiting CDK4/6 with palbociclib. Inhibition of CDK4/6 reduced the DNA-damage phenotype due to YAP5SA expression (Fig. 5J). Although S-phase entry was not completely blocked by palbociclib (Fig. EV4G), since these experiments were performed in asynchronous cell populations, the observed reduction in DNA damage could reflect a lower proportion of cells in S-phase rather than a direct effect on S-phase integrity. To address this, we investigated γH2AX levels in S-phase by high-content microscopy after restoring G1 length of YAP5SA-expressing cells with a low concentration of palbociclib, which we previously showed to prolong G1 without fully arresting S-phase entry (see Fig. 2C). These experiments confirmed reduced DNA damage in S-phase by prolonging G1 (Fig. 5K), suggesting a causal relationship between G1 shortening and the sensitizing effect of YAP towards CHK1 inhibition.

In conclusion, we identified G1 shortening as a key mechanism underlying the replication stress and sensitivity to ATR-CHK1 inhibition induced by oncogenic YAP. This G1 shortening is driven by elevated CDK4/6 activity, independent of CDK2, and results in insufficient licensing of replication origins. Unexpectedly, premature S-phase entry in YAP-expressing cells with underlicensed origins was associated with accelerated rather than slower replication fork progression. Increased fork speed and hypertranscription may lead to more frequent collisions between the replication and transcription machineries, thereby contributing to replication stress and genomic instability. Notably, the replication stress induced by YAP5SA is mild and does not lead to overt DNA damage under normal conditions but creates a specific vulnerability to CHK1 inhibition (Fig. 5L). This suggests that YAP also activates pathways to suppress replication stress. Exploring how oncogenic YAP alleviates this stress could reveal therapeutic targets for YAP-driven cancers.

## Methods

### Reagents and tools table

| Reagent/resource | Reference or source | Identifier or catalog number |
|---|---|---|
| **Experimental models** | | |
| MCF10A-YAP5SA | Björns von Eyss | |
| MCF10A-YAP5SA-PIP-FUCCI | This study | |
| MCF10A-YAP5SA- DHB-mVenus-p2a-mCherry-CDK4KTR | This study | |
| MCF10A-YAP5SA-S94A | This study | |
| MCF10A-CCNE1 | This study | |

| Reagent/resource | Reference or source | Identifier or catalog number |
|---|---|---|
| **Recombinant DNA** | | |
| GST-RHdelta-MNase | Addgene | 136292 |
| pCMV-FlagYAP5SA/S94A | Addgene | 33103 |
| pCMV-VSV-G | Addgene | 8454 |
| pInducer20 Cyclin E1 | Addgene | 109348 |
| pInducer20 YAP5SA/S94A | This study | |
| pLenti-DHB-mVenus-p2a-mCherry-CDK4KTR | Addgene | 126679 |
| pLenti-PGK-Neo-PIP-FUCCI | Addgene | 118616 |
| psPAX2 | Addgene | 12260 |
| **Antibodies** | | |
| Beta-Actin (C4) | Santa Cruz | sc-47778 |
| BrdU (B44) | BD Biosciences | 347580 |
| BrdU Proliferation Marker | Abcam | ab6326 |
| Cyclin B (H-433) | Santa Cruz | sc-752 |
| MCM7 | Santa Cruz | sc-56324 |
| PCNA | Santa Cruz | sc-56 |
| pH2A.X (Ser139) | Santa Cruz | Sc-517348 |
| pH3 (Ser10) | Millipore | 06-570 |
| pKAP1 (Ser824) | Abcam | ab70369 |
| pLAMIN A/C (Ser22) | Cell Signaling | 13448 |
| pRB (Ser807/811) (D20B12) | Cell Signaling | 8516S |
| pRPA32 (Ser33) | Bethyl (Biomol) | A300-246A |
| RB (G3-245) | BD Pharmingen | 554136 |
| RNA Pol II phospho Ser2 | Abcam | ab5095 |
| S9.6 | Millipore | # MABE1095 |
| YAP (63.7) | Santa Cruz | sc-101199 |
| Anti-mouse IgG (H + L) HRP | Thermo Fischer Scientific | G-21040 |
| Anti-ProteinA HRP | BD Biosciences | 610438 |
| Anti-mouse IgG (H + L) Alexa Fluor 568 | Thermo Fischer Scientific | A-11031 |
| Anti-mouse IgG (H + L) Alexa Fluor 488 | Thermo Fischer Scientific | A-11029 |
| Anti-mouse IgG (H + L) Alexa Fluor 594 | Thermo Fischer Scientific | A11032 |
| Anti-rabbit IgG (H + L) Alexa Fluor 568 | Thermo Fischer Scientific | A-11036 |
| Anti-rabbit IgG (H + L) Alexa Fluor 647 | Thermo Fischer Scientific | A-21245 |
| Anti-rabbit IgG (H + L) Alexa Fluor 488 | Thermo Fischer Scientific | A-11034 |
| Duolink In Situ anti-rabbit PLA Sonde PLUS | SIGMA ALDRICH | UO92002 |
| Duolink In Situ anti-mouse PLA Sonde MINUS | SIGMA ALDRICH | DUO92004 |
| **Oligonucleotides and other sequence-based reagents** | | |
| qPCR primers | This study | Table EV1 |
| Cloning primer | | |
| TGTGTCGACCGTCAGAATTGATCTACCATGGAC | SalI_FLAG_Y5SA | |
| ACATCTAGACTGAGGGCTCTATAACCATGTAAG | S94A_XbaI | |
| siRNAs | | Table EV1 |
| **Chemicals, enzymes, and other reagents** | | |
| 5-Chloro-2′-deoxyuridine (CldU) | Hycultec GmbH | HY-112669 |
| 5-Ethynyl-2′-deoxyuridine (EdU) | Jena Bioscience | CLK-N001 |
| 5-Ethynyl-uridine (EU) | Jena Bioscience | CLK-N002 |
| 5,6-dichloro-1-beta-D-ribofuranosyl-1H-benzimidazole (DRB) | Biomol | Cay10010302-50 |
| Adavosertib (AZD1775) | Hycultec GmbH | HY10993 |
| AF488-Azide | Jena Bioscience | CLK-1275 |
| AF647-Azide | Jena Bioscience | CLK-1299 |

| Reagent/resource | Reference or source | Identifier or catalog number |
|---|---|---|
| Ceralasertib (AZD6738) | Hycultec GmbH | HY19323 |
| Cholera toxin | Enzo Life Science GmbH | BML-G117 |
| DMEM/F-12 | Thermo Fisher Scientific | 31331093 |
| DMSO | SIGMA ALDRICH | 472301 |
| dNTPs | Thermo Fisher Scientific | R0186 |
| Doxorubicin-hydrochlorid | SIGMA ALDRICH | D1515 |
| Doxycycline | SIGMA ALDRICH | D9891 |
| Glutathione Sepharose 4B | VWR/GE Healthcare | 17075601 |
| Glutathione, reduced | SIGMA ALDRICH | G4251 |
| Horse serum | Thermo Fisher Scientific | 16050122 |
| Hydrocortisone | SIGMA ALDRICH | H0888 |
| Idoxuridine (IdU) | Hycultec GmbH | HY-B0307 |
| Insulin | SIGMA ALDRICH | I9278 |
| IPTG | SIGMA ALDRICH | I6758 |
| JQ1 | SIGMA ALDRICH | SML1524 |
| Nutlin-3 | SIGMA ALDRICH | 444151 |
| Page Ruler Prestained Protein Ladder | Thermo Fisher Scientific | 26617 |
| Palbociclib isothionate | Hycultec GmbH | HY-A0065 |
| Phusion DNA Polymerase | Thermo Fisher Scientific | F-530L |
| Prexasertib dihydrochloride | Hycultec GmbH | HY-18174A |
| Propidium iodide | SIGMA ALDRICH | 537059 |
| Protease Inhibitor | SIGMA ALDRICH | P8340 |
| Random primer | SIGMA ALDRICH | 11034731001 |
| Recombinant human EGF | SIGMA ALDRICH | E9644 |
| Restriction enzymes | New England Biolabs | |
| RevertAid Reverse Transcriptase | Thermo Fisher Scientific | EP0441 |
| RiboLock RNase Inhibitor | Thermo Fisher Scientific | EO0384 |
| Sodium ascorbate | SIGMA ALDRICH | A7631 |
| T4 DNA Ligase | New England Biolabs | M0202T |
| Thiazolyl blue tetrazolium bromide | SIGMA ALDRICH | M5655 |
| Trizol | Thermo Fisher Scientific | 15596018 |
| **Software** | | |
| GraphPad Prism v10 | GraphPad | |
| ImageJ 1.53k | http://imagej.net/ij/ | |
| DiffBind 3.12.0 | Ross-Innes et al (2012) | |
| MACS v2.2.9.1 | Zhang et al (2008) | |
| Picard tools 2.18.2.2 | http://broadinstitute.github.io/picard/ | |
| Bowtie v2.4.2 | Langmead et al, 2012 | |
| R 4.4.3 | https://www.r-project.org | |
| RStudio v 2024.12.1 | Posit Software, PBC | |
| Trimmomatic 0.38.1 | Bolger et al (2014) | |
| Harmony High Content Imaging and Analysis Software v4.8 | Perkin Elmer | |
| **Other** | | |
| Duolink In Situ Detection Reagents Red | SIGMA ALDRICH | DUO82008 |
| QIAquick PCR Purification Kit | Qiagen | 28106 |

| Reagent/resource | Reference or source | Identifier or catalog number |
|---|---|---|
| Quant-iT PicoGreen dsDNA Assay Kit | Thermo Fisher Scientific | P11496 |
| Illumina NextSeq 2000 P3 Reagents; 50 cycles | Illumina | 20046810 |
| Operetta™ High-Content Screening System | Perkin Elmer | |
| NextSeq 2000 sequencer | Illumina | |
| qTower3G | Analytic Jena | |

## Cell culture

MCF10A expressing doxycycline-inducible YAP5SA cells have been described before (Pattschull et al, 2019). They were cultured in DMEM/F-12 supplemented with 5% horse serum, 1% penicillin/ streptomycin, 10 µg/ml insulin, 500 ng/ml hydrocortisone, 20 ng/ml EGF and 100 ng/ml cholera toxin. Cells were treated with the indicated concentrations of doxycycline, prexasertib (CHK1i), ceralasertib (ATRi), adavosertib (WEE1i), 5,6-dichloro-1-b-D-ribofuranosylbenzimidazole (DRB), Palbociclib (CDK4/6i), doxorubicin or Nutlin-3. Drugs were obtained from Sigma.

MCF10A cells with inducible overexpression of CCNE1 and MCF10A-YAP5SA cell lines harboring CDK2 and CDK4 reporters or PIP-FUCCI were generated by lentiviral transduction. Lentiviral particles were generated in HEK293T co-transfected with psPAX2, pCMV-VSV-G and a lentiviral vector. Filtered viral supernatant was diluted 1:1 with culture medium and supplemented with 4 µg/ml polybrene (Sigma-Aldrich). Infected cells were selected 48 h after infection with the appropriate antibiotics for 7 days. The lentiviral construct encoding CDK2 and CDK4 reporters (pLenti-DHB-mVenus-p2a-mCherry-CDK4KTR) was a gift from Hee Won Yang (Addgene plasmid # 126679, RRID:Addgene_126679). pInducer20 Cyclin E1 was a gift from Jean Cook (Addgene plasmid # 109348, RRID:Addgene_109348). The PIP-FUCCI plasmid was gift from Jean Cook (Addgene plasmid # 118616, RRID:Addgene_118616). Cells were selected with blasticidin (10 µg/ml) and neomycin (300 µg/ml).

## MTT viability assays

Cells were seeded in a 96-well plate and treated the following day with the indicated concentrations of drugs in complete medium. Seventy-two hours later, 20 µl of thiazolyl blue tetrazolium bromide (5 mg/ml) was added directly to the medium and cells. After 2 h at 37 °C, 100 µl DMSO was added, the plate was incubated for 20 min at RT and the absorbance was measured in a microplate reader at 595 nm. The absorbance was normalized to the negative control of medium without cells, and the cell viability was calculated relative to the DMSO control.

## RT-qPCR

Total RNA was isolated using Trizol (Thermo Fisher Scientific). RNA was transcribed using 100 units RevertAid reverse transcriptase (Thermo Fisher Scientific). Quantitative real-time PCR reagents were from Thermo Fisher Scientific, and real-time PCR was performed using a qTower3G (Analytik Jena). Expression differences were calculated as described before (Osterloh et al, 2007). Primer sequences are listed in Table EV1.

## Immunoblotting

Cells were lysed in TNN (50 mM Tris (pH 7.5), 120 mM NaCl, 5 mM EDTA, 0.5% NP40, 10 mM $Na_4P_2O_7$, 2 mM $Na_3VO_4$, 100 mM NaF, 1 mM PMSF, 1 mM DTT, 10 mM β-glycerophosphate, protease inhibitor cocktail (Sigma)). Proteins were separated by SDS-PAGE, transferred to a PVDF membrane and detected by immunoblotting with the first and secondary antibodies.

## Immunofluorescence staining and high-content microscopy

Cells were plated in 96-well plates (Revvity, formerly Perkin Elmer). Before fixation, cells were labeled with 10 µM EdU (Sigma) for 30 min. Cells were fixed with 3% paraformaldehyde and 2% sucrose in PBS for 10 min at room temperature. Cells were permeabilized using 0.2% Triton X-100 (Sigma) in PBS for 5 min and blocked with 3% BSA in PBS-T (0.1% Triton X-100 in PBS) for 30 min. Detection of EdU-labeled DNA was performed by copper(I)-catalyzed azide-alkyne cycloaddition in 100 mM Tris pH 8.5, 4 mM CuSO4, 10 mM AFDye 488 Azide (Jena Bioscience), 10 mM L-ascorbic acid for 30 min at room temperature. Next, cells were incubated with the primary antibodies in blocking buffer overnight at 4 °C. After washing, cells were incubated with appropriate fluorophore-conjugated secondary antibody for 1 h at RT. Counterstaining for nuclei detection was performed by 2.5 mg/ml Hoechst 33342 (Sigma-Aldrich) for 10 min at RT. Images were acquired with an Operetta CLS High-Content Imaging System (Revvity, formerly Perkin Elmer) at ×20, ×40, or ×63 magnification using a water immersion objective. Images were processed and analyzed using Harmony High Content Imaging and Analysis Software (Revvity, formerly Perkin Elmer) and custom scripts in R. For the detection of chromatin-bound MCM7, cells were pre-extracted with 0.5% Triton X-100 in CSK buffer (100 mM NaCl, 300 mM sucrose, 3 mM $MgCl_2$, 10 mM HEPES-KOH (pH 7.4) for 5 min on ice, washed once with CSK buffer and then once with PBS. Afterward, cells were fixed and immunostained as described above. For the determination of CDK2 and CDK4/6 activity, MCF10-YAP5SA cells expressing DHB-mVenus and mCherry-CDK4KTR were fixed and stained for EdU and Hoechst 33342 as described above. The mean intensities of the mCherry-CDK4KTR and DHB-mVenus signals were measured in the nucleus and in a defined cytoplasmic ring region. To standardize the spatial relationship, the distance from the nuclear center to the cytoplasmic border was set to 100%, with the nuclear boundary defined at 50%. The cytoplasmic ring was delineated by setting the outer border at 25% and the inner border at 40% of this distance. CDK activity was calculated as the ratio of the mean intensity in the cytoplasmic ring region to the mean intensity in the nucleus: CDK activity = mean intensity (ring region)/mean intensity (nucleus). As previously described (Yang et al, 2020), CDK2 activity contributes to the CDK4/6 reporter signal during S/G2. To account for this, a correction factor of 0.35 was applied to the analysis, yielding a corrected CDK4/6 activity: Corrected CDK4/6 activity = CDK4/6

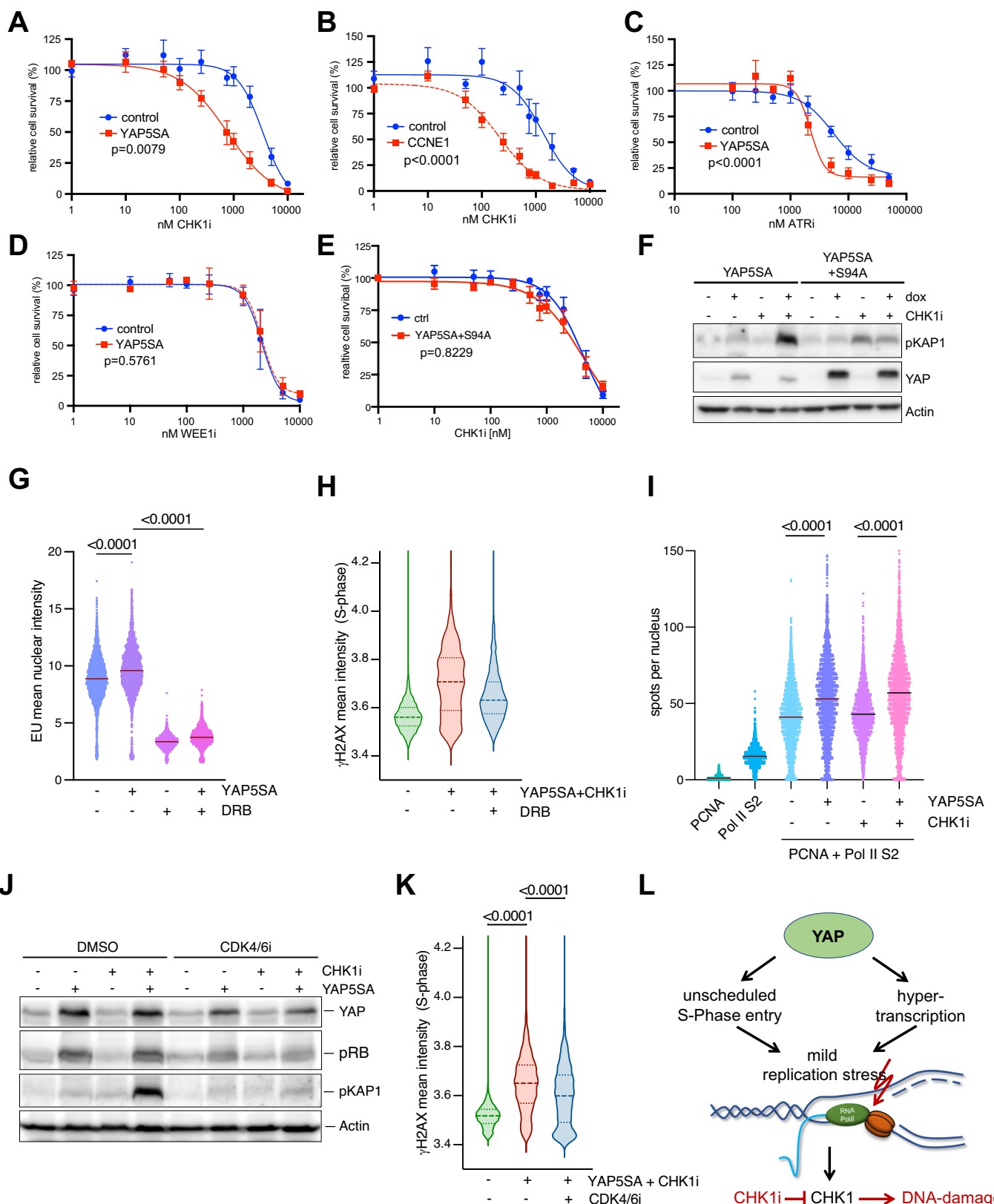

**Figure 5. Oncogenic YAP sensitizes cells to CHK1 and ATR inhibition.**

(A) Viability of cells exposed to the CHK1 inhibitor prexasertib was analyzed by MTT assay. Cells were treated for 3 days with increasing concentrations of CHK1i. (B) MTT viability assay of MCF10A cells expressing doxycycline-inducible Cyclin E1 after treatment with prexasertib. (C, D) MTT viability assay of cells exposed to the ATR inhibitor AZD6738 or the WEE1 inhibitor AZD1775. Mean $+/-$ SEM. (E) Cells expressing YAP5SA,S94A were exposed to the CHK1 inhibitor prexasertib. Viability was analyzed by MTT assays. In Fig. 5A–E data are presented as mean ± SEM. $P$ values were calculated using the extra sum-of-squares F test. (A–D) $n = 3$ independent replicates, (E) $n = 4$ independent replicates. (F) Immunoblot analysis of pKAP1 and YAP. YAP5SA and YAP5SA-S94A were expressed for 48 h and treated with 100 nM CHK1i for 8 h. Actin served as a loading control. (G) Quantification of nuclear EU intensities in S-phase after labeling for 30 min with 0.5 mM EU. The PIP-FUCCI cell cycle reporter (Fig. EV4B) was used to identify S-phase cells. Violin plots display single-cell measurements from a representative experiment, with at least 3337 cells analyzed per condition. $P$ values were calculated using ordinary one-way ANOVA ($n = 3$ independent replicates). (H) Effect of treating YAP-expressing cells with DRB on γH2AX staining. Violin plots display single-cell measurements from a representative experiment, with at least 1767 cells analyzed per condition ($n = 2$ independent replicates). (I) PLA with PCNA and RNAPII pSer2 antibodies. Quantification of nuclear PLA foci. Violin plots display single-cell measurements from a representative experiment, with at least 2323 nuclei analyzed per condition. $P$ values were calculated using ordinary one-way ANOVA ($n = 3$ independent replicates). (J) Immunoblot of YAP, pRB and pKAP1 in cells treated with 100 nM prexasertib (CHK1i, 100 nM, 8 h) and palbociclib (CDK4/6i, 500 nM, 24 h). Actin served as a loading control. (K) Cells were treated with 100 nM CHKi for 4 h and with 50 nM palbociclib for 24 h. γH2AX in S-phase in cells was determined by high-content microscopy. Violin plots display single-cell measurements from a representative experiment, with at least 2309 cells analyzed per condition. $P$ values were calculated using ordinary one-way ANOVA. ($n = 3$ independent replicates). (L) Model and summary. Source data are available online for this figure.

reporter activity $- 0.35 \times$ CDK2 reporter activity. For the detection of nascent RNA synthesis during S-phase, MCF10A-YAP5SA PIP-FUCCI cells were incubated with ethynyl uridine (EU) at a final concentration of 0.5 mM for 30 min. Where indicated, cells were treated with the transcription inhibitor 5,6-dichloro-1-ß-D-ribofuranosylbenzimidazole (DRB) 100 µM for 3 h. Cells were fixed and stained for EU and Hoechst 33342 as described above for EdU staining. S-phase cells were identified based on the PIP-FUCCI system.

## DNA fiber assays

DNA fiber assays were carried out as described before with slight modifications (Schwab and Niedzwiedz, 2011). Briefly, newly synthesized DNA was labeled by treatment with 2.5 µM 5-iodo-2-deoxyuridine (IdU; 50 µM; Sigma-Aldrich) for 20 min followed by labeling with 250 µM 5-chloro-2-deoxyuridine (CldU; 25 µM; Sigma-Aldrich) for another 20 min. Cells were collected by trypsinization and resuspended in PBS. Cells were lysed in spreading buffer (200 mM Tris (pH 7.4), 50 mM EDTA and 0.5% SDS) and DNA fibers were spread on glass slides and air-dried before fixation in a 1:3 methanol:acetic acid solution. After DNA denaturation by 2.5 M HCl for 80 min and after blocking in 5%BSA in PBS-T, CldU- and IdU-labeled tracts were detected by immunostaining using mouse anti-BrdU (B44; BD Biosciences) and rat anti-BrdU (OBT0030, Serotec) antibodies, with Alexa Fluor 568-conjugated goat anti-mouse IgG and Alexa Fluor 488-conjugated goat anti-rat IgG (Thermo Fisher Scientific) as secondary antibodies. DNA fibers were visualized with fluorescence microscopy and analyzed with ImageJ.

## Proximity ligation assay (PLA)

Cells were seeded in 384-well plates (Revvity, formerly Perkin Elmer), fixed and permeabilized with 100 µl methanol for 20 min at −20 °C and blocked with 5% BSA in PBS. Cells were incubated with the indicated mouse and rabbit antibodies in blocking buffer overnight at 4 °C. Cells were incubated with plus and minus probes directed against rabbit and mouse antibodies, respectively, for 1 h at 37 °C (Sigma-Aldrich). After washing with PLA Wash buffer A (Sigma-Aldrich), ligation was performed for 30 min at 37 °C. After washing with PLA Wash buffer A, in situ PCR amplification was performed using Detection Reagents Red (Sigma-Aldrich) for

90 min at 37 °C. Samples were counterstained with Hoechst 33342 (Sigma-Aldrich). Images were acquired using the Operetta CLS High-ContentAnalysis System at ×40 magnification (Revvity, formerly Perkin Elmer) and analyzed using Harmony High-Content Imaging and Analysis Software (Revvity, formerly Perkin Elmer) and Prism 10.

## R-loop detection by S9.6 dot blot

DNA-RNA hybrids were detected as described previously (Stork et al, 2016). Briefly, cells were harvested by trypsinization and lysed in 0.5% SDS in TE pH 8.0, with freshly added proteinase K (0.1 mg/ml) followed by phenol/chloroform extraction and EtOH/sodium acetate precipitation. Genomic DNA was resuspended in TE and spotted onto nylon membranes (Amersham Hybond-N, GE Healthcare, Cat# RPN303N). The membrane was subjected to UV crosslinking (120 mJ/cm$^2$) and then blocked and probed with S9.6 antibody (1:500, Millipore, Cat# MABE1095) overnight at 4 °C. Detection was done with an HRP-conjugated secondary antibody and blot development with ECL reagents. After S9.6 signal detection, the membrane was washed and stained with methylene blue (0.02% in 0.5 M sodium acetate, pH 5.2) for 30 min at room temperature to detect total genomic DNA. The S9.6 and methylene blue signal were quantified using ImageJ; the S9.6 values were normalized to the methylene blue signal values.

## Purification of GST-ΔRH-Mnase

A catalytically inactive RNase H fused to micrococcal nuclease (MNase), GST-ΔRH-MNase, was purified as previously described (Yan and Sarma, 2020). Briefly, *E. coli* BL21 transformed with pGEX-6P-1 GST-RH-MNase (Addgene, # 136292) were grown overnight, and protein expression was induced by adding 1 mM IPTG for 3 h. Bacterial cells were lysed by sonication, and the cleared lysate was incubated with glutathione sepharose beads overnight at 4 °C. After extensive washing, GST-ΔRH-MNase was eluted using GST elution buffer (125 mM Tris-HCl pH 8.0, 150 mM NaCl, 10 mM reduced L-glutathione) and subsequently dialyzed against BC100 buffer (50 mM Tris-HCl pH 7.6, 2 mM EDTA, 100 mM KCl, 10% (v/v) glycerol, 0.1 mM DTT, 0.2 mM PMSF). Protein concentration and purity were assessed by Bradford assay and SDS-PAGE, followed by Coomassie staining.

## MapR

For MapR experiments, MCF10A-YAP5SA cells were left untreated or treated with 0.5 μg/ml doxycycline for 48 h, 100 nM CHK1 inhibitor for 8 h, or 500 nM JQ1 for 16 h. MapR was performed as previously described (Yan et al, 2019; Yan and Sarma, 2020). Briefly, concanavalin A-coated magnetic beads were activated in binding buffer (20 mM HEPES-KOH pH 7.9, 10 mM KCl, 1 mM $CaCl_2$, 1 mM $MnCl_2$). Cells were trypsinized, washed with wash buffer (20 mM HEPES–NaOH pH 7.5, 150 mM NaCl, 0.5 mM spermidine, protease inhibitor), and counted. Cells were incubated with the activated beads in 1 ml wash buffer for 1 h at room temperature. After discarding the supernatant, cells were resuspended in 50 μl wash buffer containing 0.005% digitonin and incubated overnight at 4 °C with 1 μM GST-ΔRH-MNase. MNase was activated by adding 2 mM $CaCl_2$ and incubating for 30 min at 0 °C. The reaction was stopped by adding 2× stop buffer (340 mM NaCl, 20 mM EDTA, 4 mM EGTA, 0.005% digitonin, 0.05 mg/ml RNase A, 0.05 mg/ml linear acrylamide). To release chromatin fragments, samples were incubated at 37 °C for 10 min, followed by incubation at 70 °C in the presence of 0.1% SDS and 5 μg proteinase K. DNA was purified by phenol–chloroform extraction and processed for library preparation using the NEBNext Ultra II DNA Library Prep Kit for Illumina (NEB), followed by sequencing on an Illumina NextSeq 2000 platform.

Sequencing reads were adapter-trimmed using Trimmomatic (Bolger et al, 2014), and aligned to the human reference genome (hg38) using Bowtie2 (Langmead and Salzberg, 2012). To identify changes in R-loop regions, broad peaks were called using MACS2 ($P$ value < 0.05) (Zhang et al, 2008), and R-loop enrichment across two independent replicates was quantified using DiffBind (Ross-Innes et al, 2012).

## siRNA transfection

Double-stranded RNA was purchased from Eurofins or Thermo Fischer Scientific. siRNAs were transfected using RNAiMAX (Thermo Fisher Scientific). siRNAs are listed in Table EV1.

## Flow cytometry

Samples were washed with ice-cold PBS and fixed in 80% ice-cold ethanol. Then, cells were washed with ice-cold PBS and resuspended with 38 mM sodium citrate with 500 μg/ml RNase A for 30 min at 37 °C. Cells were stained 43 mM propidium iodide and cells were analyzed on a BD FACS Diva.

## Statistical analysis

Statistical methods and the number of replicates is reported in the Figure legend. Statistical analyses were performed using Prism 10 (GraphPad) and R. Exact $P$ values are provided for values >0.0001. Below 0.0001, $P$ values are provided as "<0.0001". No statistical methods were used to predetermine sample size. Fiber assay samples were assigned numerical labels, randomized and analyzed in a blinded manner to minimize downstream operator bias. No randomization or blinding was applied to other experiments.

## Graphics

Model images were created with BioRender.com.

## Data availability

MapR datasets produced in this study are available at Gene Expression Omnibus (GEO) under accession number GSE296501.

The source data of this paper are collected in the following database record: biostudies:S-SCDT-10_1038-S44319-025-00514-5.

## Peer review information

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

## Acknowledgements

The authors thank Franziska Schönhofer, Marco Jessen and Ursula Eilers for assistance with cell proliferation experiments, dot blot experiments and high throughput microscopy. This work was supported by grants from the Deutsche Krebshilfe (70112811) and by the Deutsche Forschungsgemeinschaft (DFG, German Research Foundation) GA 575/10-1, GA 575/10-2 and 440766788 (INST 93/1023-1-FUGG).

## Author contributions

**Dörthe Gertzmann**: Conceptualization; Software; Investigation; Visualization; Methodology; Writing—review and editing. **Cornelius Presek**: Investigation; Visualization. **Anna Lena Mattes**: Investigation; Visualization. **Marco Sänger**: Investigation; Visualization. **Marie Zoller**: Investigation. **Christina Schülein-Völk**: Resources; Investigation. **Carsten P Ade**: Resources; Investigation. **Martin Eilers**: Resources; Funding acquisition. **Stefan Gaubatz**: Conceptualization; Supervision; Investigation; Visualization; Writing—original draft; Writing—review and editing.

Source data underlying figure panels in this paper may have individual authorship assigned. Where available, figure panel/source data authorship is listed in the following database record: biostudies:S-SCDT-10_1038-S44319-025-00514-5.

## Funding

## Disclosure and competing interests statement

The authors declare no competing interests.

# Expanded View Figures

**Figure EV1. Oncogenic YAP activates CDK4/CDK6 in G1, but does not lead to CDK2 activation.**

(A) Representative immunofluorescence images showing the activity-sensors for CDK2 (DHB-mVenus) and CDK4 (mCherry-CDK4KTR). Scale bar: 100 μm. (B) CDK4/6 and CDK2 activity during S and G2 phase in control and YAP5SA-expressing MCF10A cells. Violin plots display single-cell measurements from at least 2064 cells per condition. *P* values were calculated using unpaired Student's *t* test ($n = 3$ independent replicates). (C) CDK2 activity of control and YAP5SA-expressing MCF10A cells treated with increasing concentrations of palbociclib, measured by high-content microscopy using the CDK2 reporter construct. Data represent single-cell analysis from at least 1038 cells per condition from a representative experiment ($n = 3$ independent replicates). (D) Distribution of the pRB/RB ratio in S-phase. Violin plots display single-cell measurements from a representative experiment, with at least 1441 cells analyzed per condition. *P* values were calculated using ordinary one-way ANOVA ($n = 3$ independent replicates). (E) Density plots of the pRB/RB ratio for the conditions indicated in the legend. Kernel density estimation (KDE) was used to visualize the distribution of the pRB/RB ratios without binning. The density is normalized such that the area under each curve equals 1. (F) RT-qPCR analysis of the indicated G1 cyclins and CDKs. Data are shown as mean ± SD. Statistical significance was assessed using Student's *t* test ($n = 4$ independent replicates). (G) Western blot analysis of the indicated G1 cyclins and CDKs. Actin served as a loading control. Source data are available online for this figure.

▶

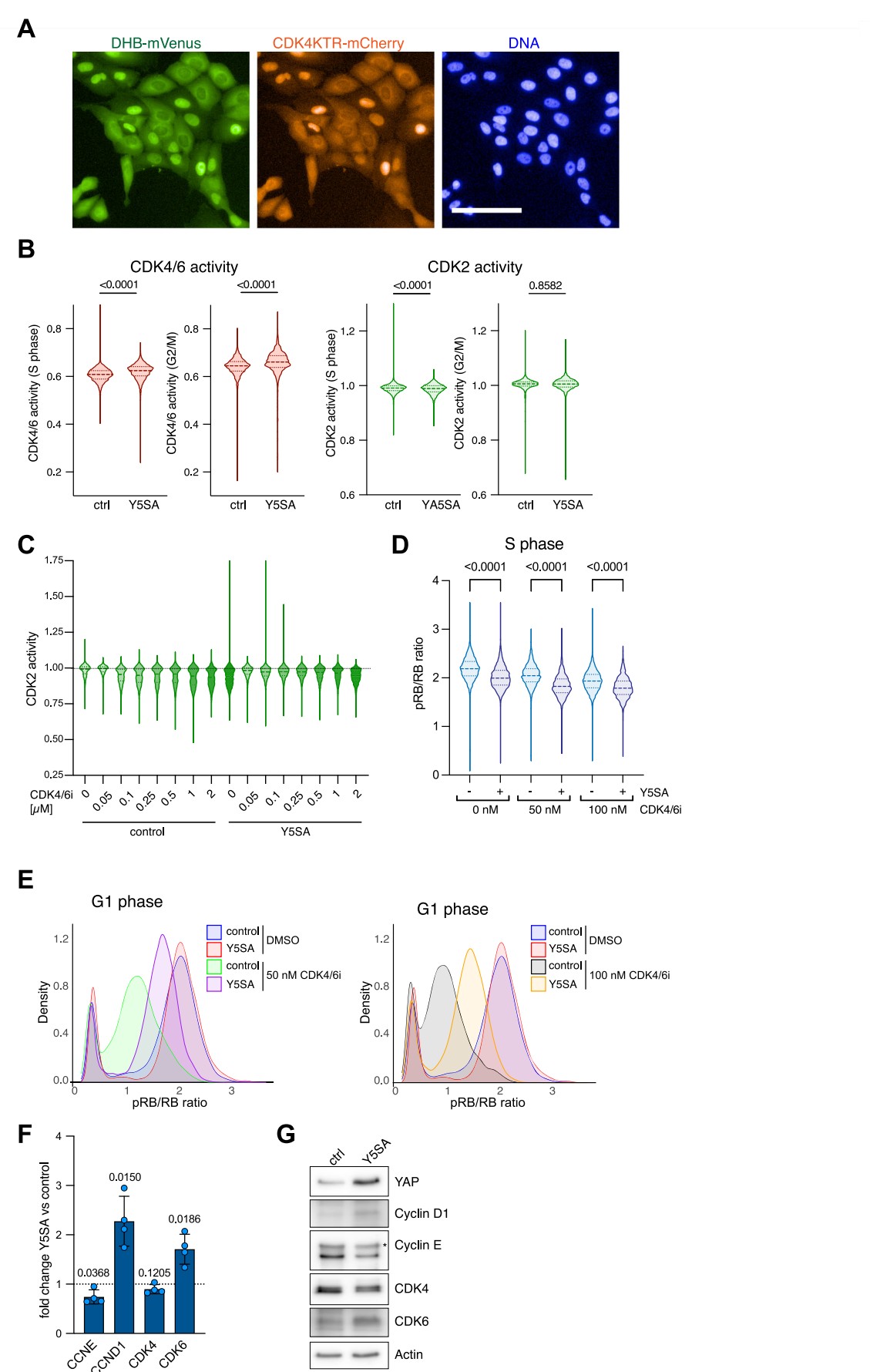

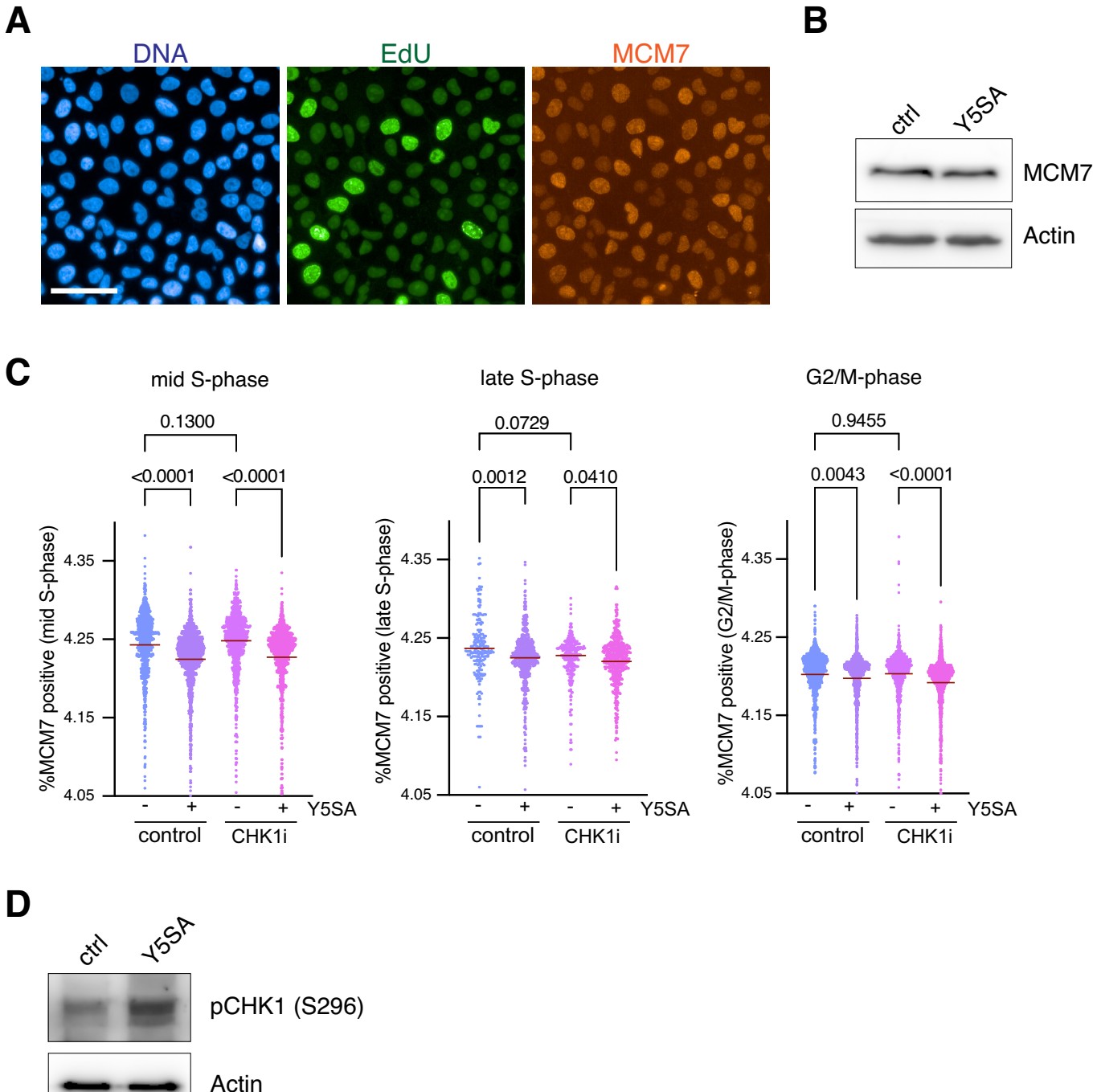

**Figure EV2. YAP5SA does not alter MCM7 protein levels.**

(A) Representative immunofluorescence image of MCM7 staining. Scale bar: 50 μm. (B) Immunoblot analysis to determine MCM7 levels. Actin served as a loading control. (C) Chromatin-bound MCM7 in mid and late S-phase and in G2. Violin plots display single-cell measurements from a representative experiment, with ≥204 cells analyzed per condition. P values were calculated using ordinary one-way ANOVA ($n = 3$ independent replicates). (D) Immunoblot analysis to determine pCHK1 (S296). Actin served as loading control. Source data are available online for this figure.

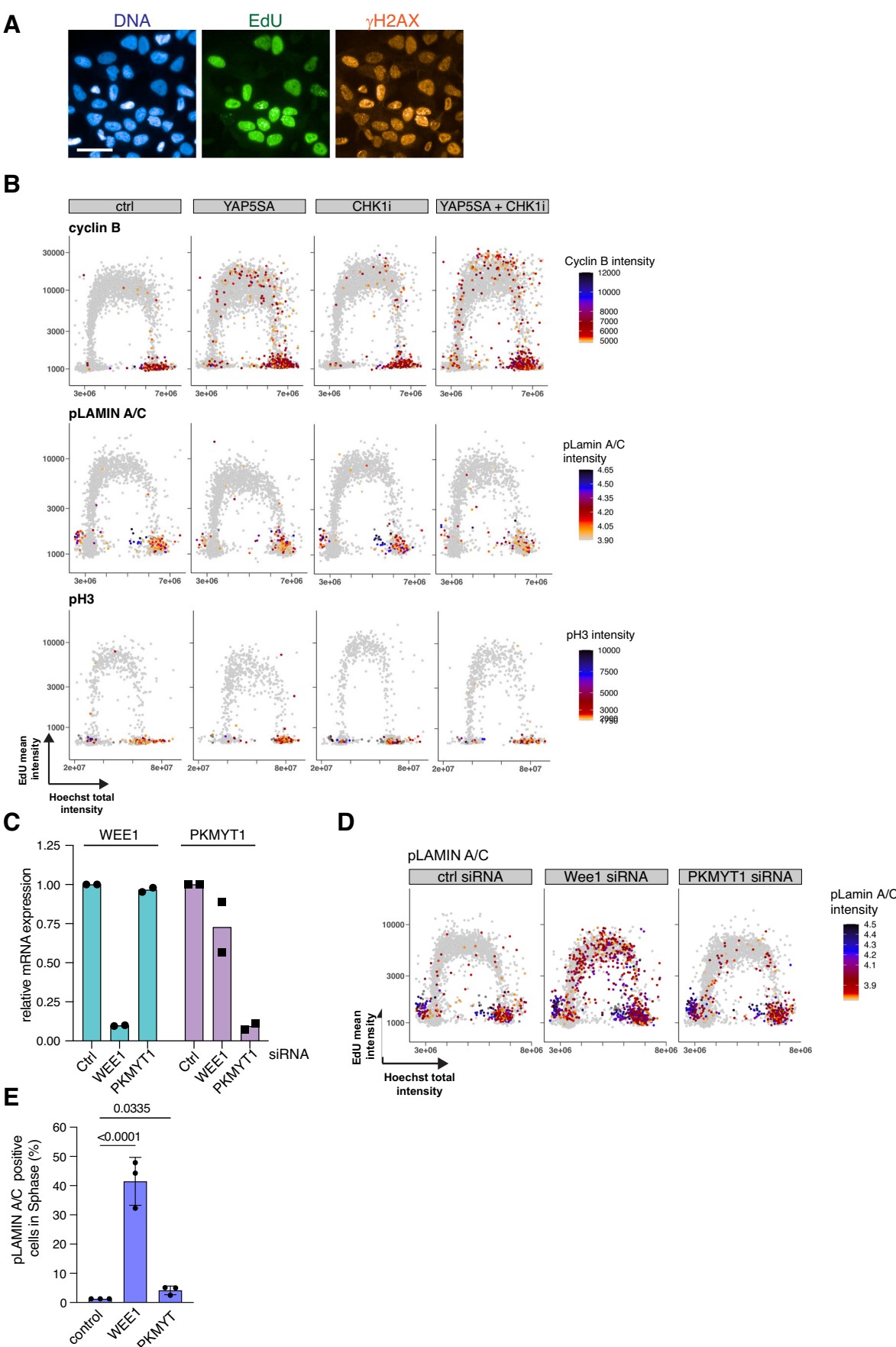

**Figure EV3.  CHK1 inhibition in YAP5SA-expressing cells does not lead to premature mitosis in S-phase.**

(A) Representative immunofluorescence images of γH2AX immunostaining. Scale bar: 50 µm. See also Fig. 4B. (B) High-content microscopy-based analysis of cytoplasmic cyclin B, pLAMIN A/C(S22) and pH3(S10). S-phase cells were labeled with EdU. For each panel, the following number of cells were randomly selected: cyclin B: 5000 cells, pLamin: 3500 cells, pH3: 1600 cells ($n = 3$ independent replicates). (C) RT-qPCR was used to validate the siRNA-mediated knockdown of WEE1 and PKMYT1 ($n = 2$ independent replicates). (D) High-content microscopy-based analysis of pLAMIN A/C(S22) in cells transfected with WEE1 and PKYT1 specific siRNAs or with a control siRNA. For each sample, 4000 cells were randomly selected ($n = 3$). (E) Quantification of the percentage of cells positive for pLAMIN A/C(S22) in S-Phase in the experiment shown in (D). Mean $+/-$ SD. $P$ values were calculated using ordinary one-way ANOVA ($n = 3$ independent replicates). Source data are available online for this figure.

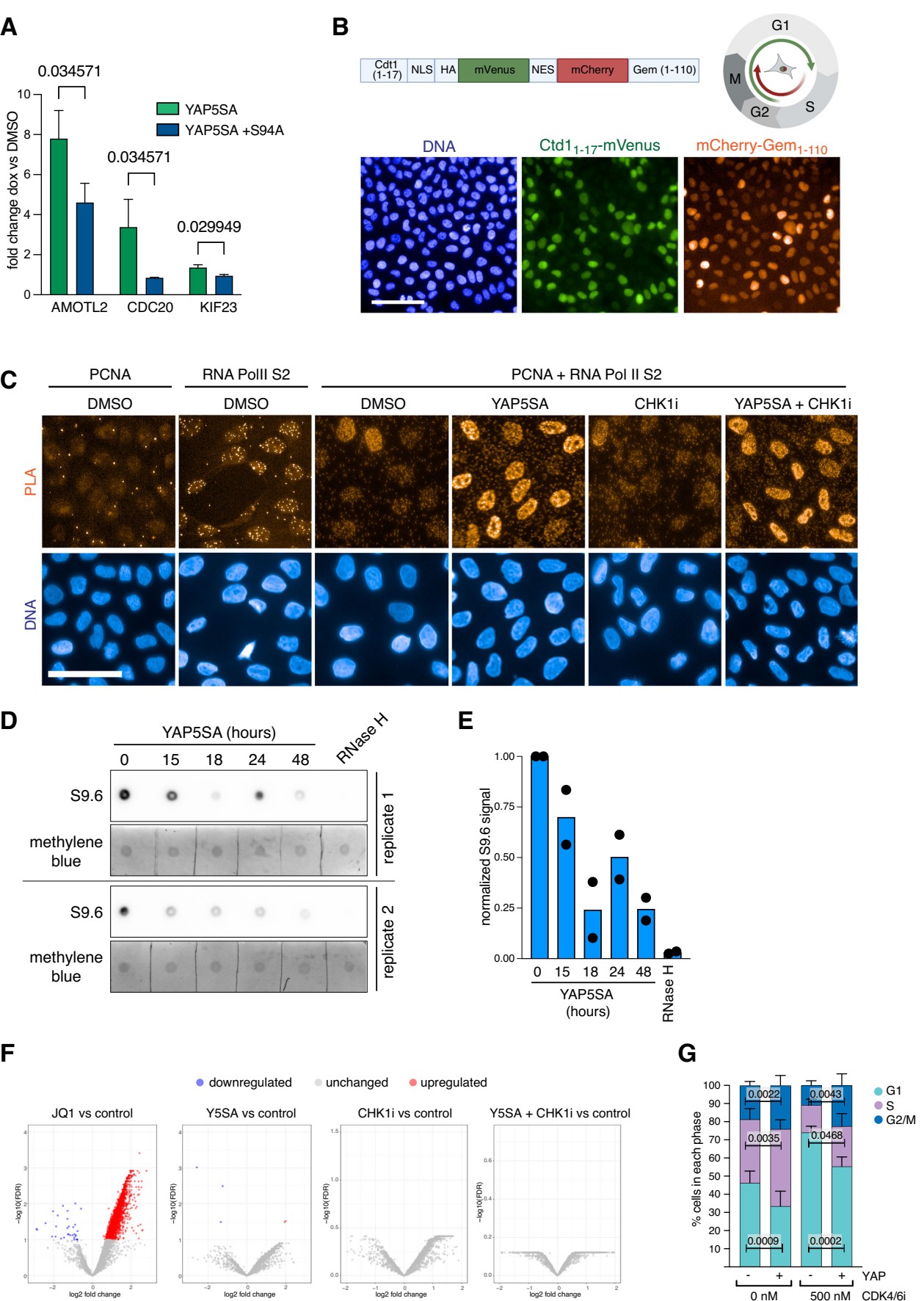

**Figure EV4.  YAP5SA enhances the proximity between the transcription and replication machinery but does not lead to increased R-loop levels.**

(A) RT-qPCR was used to analyze expression of the indicated genes upon expression of YAP5SA or YAP5SA-S94A. Mean $+/-$ SD. P values were calculated using an unpaired Student's $t$ test ($n = 3$ independent replicates). (B) Scheme of the PIP-FUCCI cell cycle sensors and representative images of MCF10A-YAP5SA cells expressing the sensors. Scale bar: 100 μm. (C) Representative images of Proximity Ligation assay (PLA) between PCNA and RNA Pol II phosphorylated at Serine 2. Scale bar: 50 μm. See Fig. 5I. (D) Dot blot assay to detect R-loops in control MCF10A cells and cells expressing YAP5SA for the indicated times. Total DNA was analyzed with the S9.6 monoclonal antibody, which recognizes DNA:RNA hybrids. RNase H1 treatment served as a control. Methylene blue staining was used as a control for equal loading ($n = 2$ independent replicates). (E) Quantification of the S9.6 dot blot signal in (D) normalized to total DNA detected by methylene blue staining ($n = 2$ independent replicates). (F) MapR was used to assess R-loop levels in control MCF10A cells and YAP5SA-expressing cells control treated or after treatment with CHK1i. Treatment with the BRD4 inhibitor JQ1, which has been shown to increase R-loop levels, served as positive control. MapR peaks were called by MACS2. The volcano plot shows differential R-loops as identified by DiffBind ($n = 2$ independent replicates). (G) Fraction of cells in each phase of the cell cycle with and without treatment with 500 nM palbociclib for 24 h. Mean $+/-$ SD. P values were calculated using ordinary one-way ANOVA ($n = 5$ independent replicates). Source data are available online for this figure.

