## [Peer Review File · EMBO Reports]

Oncogenic YAP sensitizes cells to CHK1 inhibition via CDK4/6 driven G1 acceleration

Dörthe Gertzmann, Cornelius Presek, Anna Mattes, Marco Sängler, Marie Zoller, Christina Schüle-Völk, Carsten Ade, Martin Eilers, and Stefan Gaubatz

Corresponding author(s): Stefan Gaubatz (stefan.gaubatz@biozentrum.uni-wuerzburg.de)

Review Timeline:

Submission Date:	11th Feb 25
Editorial Decision:	11th Mar 25
Revision Received:	22nd May 25
Editorial Decision:	10th Jun 25
Revision Received:	11th Jun 25
Accepted:	20th Jun 25

Editor: Esther Schnapp

Transaction Report:

Dear Dr. Gaubatz,

Thank you for the submission of your manuscript to EMBO reports. We have now received the full set of referee reports that is pasted below.

As you will see, the referees acknowledge that the findings are potentially interesting. However, they also have several suggestions for how the study should be strengthened. Apart from adding more experimental data, statistics need to be provided and the findings need to be better placed in the context of the literature. I think all referee suggestions are good and should be addressed. Please let me know in case you disagree and we can discuss the exact revision requirements further, also in a video chat, if you wish.

I would thus like to invite you to revise your manuscript with the understanding that the referee concerns must be fully addressed and their suggestions taken on board. Please address all referee concerns in a complete point-by-point response. Acceptance of the manuscript will depend on a positive outcome of a second round of review. It is EMBO reports policy to allow a single round of major revision only and acceptance or rejection of the manuscript will therefore depend on the completeness of your responses included in the next, final version of the manuscript.

We realize that it is difficult to revise to a specific deadline. In the interest of protecting the conceptual advance provided by the work, we recommend a revision within 3 months (11th Jun 2025). Please discuss the revision progress ahead of this time with the editor if you require more time to complete the revisions.

- 1) A data availability section providing access to data deposited in public databases is missing. If you have not deposited any data, please add a sentence to the data availability section that explains that.
- 2) Your manuscript contains statistics and error bars based on $n=2$. Please use scatter blots in these cases. No statistics should be calculated if $n=2$.

3) We replaced Supplementary Information with Expanded View (EV) Figures and Tables that are collapsible/expandable online. A maximum of 5 EV Figures can be typeset. EV Figures should be cited as 'Figure EV1, Figure EV2' etc... in the text and their respective legends should be included in the main text after the legends of regular figures.

5) a complete author checklist, which you can download from our author guidelines . Please insert information in the checklist that is also reflected in the manuscript. The completed author checklist will also be part of the RPF.

6) Please note that all corresponding authors are required to supply an ORCID ID for their name upon submission of a revised manuscript (). Please find instructions on how to link your ORCID ID to your account in our manuscript tracking system in our Author guidelines

7) Before submitting your revision, primary datasets produced in this study need to be deposited in an appropriate public database (see <https://www.embopress.org/page/journal/14693178/authorguide#datadeposition>). Please remember to provide a reviewer password if the datasets are not yet public. The accession numbers and database should be listed in a formal "Data Availability" section placed after Materials & Method (see also <https://www.embopress.org/page/journal/14693178/authorguide#datadeposition>). Please note that the Data Availability Section is restricted to new primary data that are part of this study. * Note - All links should resolve to a page where the data can be accessed. *
If your study has not produced novel datasets, please mention this fact in the Data Availability Section.

- the name of the statistical test used to generate error bars and P values,
- the number (n) of independent experiments (please specify technical or biological replicates) underlying each data point,
- the nature of the bars and error bars (s.d., s.e.m.),
- If the data are obtained from n {less than or equal to} 2, use scatter blots showing the individual data points.

12) All Materials and Methods need to be described in the main text using our 'Structured Methods' format, which is required for all research articles. According to this format, the Methods section includes a Reagents and Tools Table (listing key reagents, experimental models, software and relevant equipment and including their sources and relevant identifiers) followed by a Methods and Protocols section describing the methods using a step-by-step protocol format. The aim is to facilitate adoption of the methodologies across labs. More information on how to adhere to this format as well as a downloadable template (.docx) for the Reagents and Tools Table can be found in our author guidelines:
<https://www.embopress.org/page/journal/14693178/authorguide#structuredmethods>.

An example of a Method paper with Structured Methods can be found here: <https://www.embopress.org/doi/full/10.1038/s44320-024-00037-6#sec-4>

I look forward to seeing a revised form of your manuscript when it is ready.

Referee #1:

Dörthe Gertzmann and coworkers have studied the effects of oncogenic YAP on the S-phase entry and progression using MCF10A cells expressing a doxycycline-inducible allele of YAP carrying an activating S5A mutation. They show that oncogenic YAP shortens G1 phase via CDK4/6, which leads to replication stress and enhanced sensitivity of cells to pharmacological inhibition of the intra-S-phase checkpoint.

Overall, this is a well-written manuscript that 1) elucidates how YAP hyperactivation can cause DNA replication stress, and 2) exposes potential therapeutic vulnerability to CHK1 (or ATR) inhibitors. These findings are novel and of general interest to the cell cycle and tumor biology fields, because Hippo/YAP signaling is frequently deregulated in cancer, and understanding the oncogenic actions of YAP may contribute to development for personalized therapeutic strategies. I have a few concerns with regard to the experimental approaches, and some suggestions to further strengthen the manuscript.

Major comments

1. The authors use EU pulsing to quantify nascent RNA, to quantify overall transcription rates. They conclude that oncogenic YAP causes hypertranscription. However, the results may be biased by the fact the fraction of cycling cells is increased in YAP-expressing cells, and the notion that cell cycle stage may affect global transcription rates. It would be better to measure EU signals specifically in S-phase cells. This can be solved by double-pulsing and co-staining the cells with IdU and EU.
2. Proximity ligation assays are used to detect transcription-replication conflicts (TRCs). But detection of close proximity between PCNA and Pol2-phosphoSer2 does not necessarily prove that there are TRCs, if overall transcription is increased by YAP. I have no suggestion for complementary experiments to support this claim other than looking at DNA:RNA hybrids using the S9.6 antibody. This is what the authors have already done with slot blots, but the results were contrary to what one would expect, and the experiment is not conclusive in its present form. They should normalize the blot signals for total DNA content using a dsDNA antibody. Furthermore, a quantification of multiple replicates instead of just showing one example experiment would be more convincing.
3. Following up on the above point, it is unclear why RNA:DNA hybrids seem reduced. Could upregulation of RNase H2 expression by YAP offer a simple mechanistic explanation? It should be quite straightforward to test this.
4. In the description of figure 5K and 5L the authors claim that "inhibition of CDK4/6 activity and restoration of G1 length by palbociclib partially reduced the DNA-damage phenotype due to YAP expression". However they do not provide proof that G1 length was actually restored by palbociclib. It seems to me that the palbociclib dosage is crucial in this experiment. If they blocked S-phase entry altogether in at least a subset of the cells with a high dose of palbociclib, they may also see lower DNA damage, simply because they arrest more cells in S-phase. I could not find what dosage they used in this experiment, so that should be mentioned in the figure (legend). But they should also show that average G1 duration is prolonged by palbociclib in this experimental setup, using flow cytometry or microscopy.

Minor comments

5. Following up on point 1: previous work also suggests that hyperactivation of YAP/TAZ causes replication stress and hypertranscription in neural progenitor cells (PMID:30523785). The authors should cite this paper, and place their work in the light of this existing literature, for example in the Discussion section.
6. Page 7 "This reduction in EdU... CHK1 activity [36-38]. The word "was" or "is" seems to be omitted after the word "which".
7. Page 12 "The presence... replication fork to stall." The word "of" seems to be omitted after "presence".

Referee #2:

In this manuscript, Gertzmann et al. overexpress a YAP oncogenic variant and investigate its consequences on the replicative

program. The authors show that cells expressing YAP5SA proliferate faster due to a hyperactivation of Cdk4/6, which shortens G1. They observe that YAP5SA-expressing cells suffer from origin underlicensing, which leads to a faster fork speed. This effect can be counteracted with Chk1 inhibition, at the expenses of accumulating DNA damage in YAP5SA expressing cells, becoming more sensitive to long treatments of Chk1i, but also ATRi. Finally inhibition of transcription in the YAP5SA-expressing cells treated with Chk1i, seem to decrease DNA damage suggesting a role for transcription-replication conflicts, albeit they find increased proximity of PCNA and elongating RNA Pol II in this context.

The manuscript is well written, but certain aspects could be modified to be clearer and more cautious with conclusions. The concepts of oncogenes having an impact on DNA replication and the replication stress response is not completely novel, but the mechanism of YAP seems to be different than those for myc or cyclin E. I believe this manuscript introduces key insight on the mechanism, which could be further complemented or clarified before publication by addressing the following concerns.

Major:

- In figure 2C no statistical comparison is provided, and I believe there is a noticeable accumulation of cells in G2/M for the YAP5SA-overexpressing cells, that might be significant. How do the authors explain this accumulation? Provide statistics and possible explanation in the text.
- How does YAP5SA promote higher cyclin D-CDK4/6 activation? Is it at the protein levels of any of these components? Or does it have to do with an indirect effect through a different pathway regulating CDK4/6 activity? It would be a key point to add to Figure 2.
- Figure 2H, no error bars or statistical significance is shown in the figure. Please provide both in order to be able to conclude.
- In Figure 3A authors provide graphs of MCM7 intensity in early S phase. They establish tight thresholds to show MCM7+ cells but I believe some MCM7 levels should be present at late S-phase. I would suggest to re-analyze and complete this experiment.
1) Since G1 is the cell cycle stage where replication origin licensing finishes, I would show quantification of MCM7 levels in EdU-/low Hoechst (G1). Then provide QIBC plots as provided but setting the lower threshold of MCM7 to EdU- high Hoechst (G2) where eventually all MCM complexes should have been unloaded. Overall quantifications of G1, S and G2 could be provided; and then MCM7 levels in all S-phase cells could be represented as they have in Figure 3A. Further subdivision of the S-phase in early, mid, and late could be useful as well to understand MCM unloading dynamics.
- In figure 3B, the % of positive cells would always depend on a threshold established by the authors, I would suggest plotting intensities of individual cells in a similar plot to the EdU intensity as in Figure 3C.
- As already reported in literature (PMID: 29959228) origin firing and fork speed are interconnected. Probably the less availability of origins in YAP5SA-overexpressing cells makes faster forks. On the other hand, unrestrained origin firing in Chk1 inhibited cells, makes slower forks, counteracting the low origin availability in YAP5SA-overexpressing cells. This should be mentioned in the results or in the discussion.
- The proximity between PCNA and pRNA Pol II is being used as an indicative of TRC, but in fact, literature is full of examples where this does not correlate (PMID: 38581679, opposite trends in terms of TRC than that expected for the phenotype; PMID: 37468626, RNA Pol II association with replication outside the context of TRCs) as in the case of YAP5SA overexpressing cells. This is also in line with the absence of asymmetry, as observed in Figure 3H. Therefore, the fact that these cells suffer from TRCs is a statement that should be toned down to a "higher proximity between transcription and replication machineries".
- In Supplementary Figure S5, the S9.6 could benefit of a control dot blot with the other half of the sample where the total amount of loaded DNA is shown for all conditions, as detected by anti-dsDNA (PMID: 36864174). Quantifications could then be performed using this control as reference.
- Since YAP5SA-overexpressing cells have DNA damage only in combination with Chk1 inhibition, does Chk1i trigger an increase in R-loops (dot blot) and TRC (PCNA:pRNA Pol II PLA) in YAP5SA-overexpressing cells?
- Figure 5K is a WB of the whole cell population, thus upon treatment with Palbociclib cells accumulate in the G1, hence less S-phase cells and less signaling of DNA damage. This should be indicated in the text. The conclusion does not suffer since Figure 5L nicely shows a decrease in the S-phase.
- Could the sensitivity of YAP5SA-overexpressing cells to Chk1i and ATRi in terms of viability be rescued by CDK4/6 inhibition?
- Previous literature on the role of YAP in DNA replication in different organisms is not mentioned e.g. PMID: 35838349, which proposes a role for YAP together with RIF1 to regulate DNA replication program in *Xenopus*. I believe this should be included in the text in the Introduction section. Moreover, how much of the effects observed in YAP5SA has to do with its interaction with RIF. Depletion of RIF1 might be tested upon expression of YAP5SA in key experiments to complement the story and provide further mechanistic detail.

Minor:

- YAP is defined as "Yes-associated protein" in page 4 while it has been introduced before either abstract or first word of the introduction.
- First title in Results, I believe authors meant "Oncogenic" instead of "Oncogenes".
- Cytoplasmic endogenous YAP is barely visible in Fig 1A, please provide a different image where this is properly shown.
- I would suggest changing "YAP-expressing/overexpressing cells" by "YAP5SA-overexpressing cells" to avoid confusion. Same for the labeling of the figures.
- Figure Legend 1. 10 nM EdU (I believe this is a mistake and it should be 10 μ M) while in methods it states "treated with 10 mM EdU". I recommend stating "labeled instead of treated" and write the final concentration (10 μ M?) not the stock solution (10 mM?)
- Representative images of both reporters might be shown in Figure 2B, at least for the first time it is used in the manuscript. This should be done as well for Fig. 3A, 4B and 5J.
- Page 6: when mentioning "(Figure 2K,L)" add spacing before L and full stop "." right after parenthesis.
- Rephrase 3rd title of Results.
- The authors show a dependency on CHK1 kinase activity to restrain EdU incorporation. Do YAP5SA-overexpressing cells have a higher activation Chk1 kinase?
- A bit more detail in the text on how DNA fibers are labelled and produced (before mentioning Figure 3F) could be helpful for non-experts.
- Typo in Figure 4D "y axis".
- In figure 4G, I would not state that damaged cells are "stalled in mid replication" since EdU is being incorporated albeit at lower levels. This phenotype is more indicative of a severe replication defect than a complete stall.
- Third lane of "Oncogenic YAP sensitizes cells to CHK1 and ATR inhibition" section, there is Chk1 "inhibitor" missing.
- The % of gammaH2AX positive cells in S-phase in Figure 5I could be substituted for a plot of gamma intensity in these cells avoiding establishing arbitrary thresholds.

Referee #3:

Gertzmann et al. present compelling data on the role of YAP in modulating cell cycle progression and its implications for replication stress and CHK1 inhibitor sensitivity. The findings contribute valuable insights into how YAP-driven cancers may be targeted therapeutically. However, there are a few concerns and clarifications needed to strengthen the study.

Specific Comments:

1. CDK4/6 activity calculation: Given that KTR-based sensors were designed using the CDK2 sensor, a correction factor needs to be applied to the CDK4/6 sensor. For MCF-10A cells, the correction factor has been previously calculated. Corrected CDK4/6 activity = CDK4/6 reporter activity - 0.35*CDK2 reporter activity.
2. Figure 2B-CDK2 Activity and YAP Overexpression: Since CDK2 activity facilitates the G1/S transition, it is important to evaluate its dynamics throughout the cell cycle. If Figure 2B measurements focus only on G1-phase cells, the potential for no observed differences in CDK2 activity despite YAP overexpression remains. A more detailed analysis of CDK2 activity across different phases, particularly in the context of YAP overexpression, would enhance the conclusions.
3. Figure 2E - Rb Phosphorylation Interpretation: Rb phosphorylation can be maintained in S-phase cells due to the restriction point. Therefore, S-phase cells should retain Rb phosphorylation regardless of CDK4/6i treatment. Do they indeed maintain Rb phosphorylation, but the distinction between phosphorylated and non-phosphorylated populations becomes less clear after CDK4/6i treatment? Additionally, does YAP overexpression further enhance this separation in the presence of CDK4/6i?

Clarifying these points would help in interpreting the results more effectively.

4. Figure 2F-L - the potential impact of YAP overexpression on CDK2 activity: These results also be explained by the potential of CDK2 activity promotion by YAP overexpression.

5. Figure 5G - Potential Enhancement of E2F and CDK2 Activity: YAP overexpression may increase global mRNA transcription and E2F activity, thereby promoting CDK2 activation and facilitating the G1/S transition

Minor comments:

1. Figure 4D y-axis Label Correction: "Cytoplanic" should be corrected to "cytoplasmic."

2. Scatter plots (e.g. Figure 4B): The number of plotted cells should be consistent across different conditions to allow for direct comparisons. Additionally, the total number of cells analyzed should be indicated in the figure legend.

Overall, this study presents significant and well-supported findings. Addressing these points will further enhance the manuscript's clarity and impact.

EMBOR-2025-61320V1

We would like to thank the reviewers for their constructive comments on our work. The response to the comments made by the reviewers and changes to the manuscript are outlined below:

Referee #1

1. The authors use EU pulsing to quantify nascent RNA, to quantify overall transcription rates. They conclude that oncogenic YAP causes hypertranscription. However, the results may be biased by the fact the fraction of cycling cells is increased in YAP-expressing cells, and the notion that cell cycle stage may affect global transcription rates. It would be better to measure EU signals specifically in S-phase cells. This can be solved by double-pulsing and co-staining the cells with IdU and EU.

To address the concern regarding potential cell cycle-dependent bias in EU-incorporation, we quantified the EU signal in S-phase. To do so we generated MCF10A-YAP5SA cells stably expressing the PIP-FUCCI reporter, an improved version of the FUCCI system described by Grant et al. (2018), which allows for identification of S-phase cells. We first validated by EdU labeling that S-phase cells can be identified with the PIP-FUCCI reporter by the absence of green fluorescence. We then pulsed PIP-FUCCI cells with EU, followed by fixation and detection of EU incorporation. Quantification of EU intensities confirms that YAP5SA expression leads to increased nascent RNA synthesis within S-phase cells (see Figure 5G)

2. Proximity ligation assays are used to detect transcription-replication conflicts (TRCs). But detection of close proximity between PCNA and Pol2-phosphoSer2 does not necessarily prove that there are TRCs, if overall transcription is increased by YAP. I have no suggestion for complementary experiments to support this claim other than looking at DNA:RNA hybrids using the S9.6 antibody. This is what the authors have already done with slot blots, but the results were contrary to what one would expect, and the experiment is not conclusive in its present form. They should normalize the blot signals for total DNA content using a dsDNA antibody. Furthermore, a quantification of multiple replicates instead of just showing one example experiment would be more convincing.

We thank the reviewer for this important comment. We agree that proximity ligation assays (PLAs) between PCNA and phospho-Ser2 Pol II do not definitively demonstrate transcription-replication conflicts (TRCs), especially in the context of increased global transcription. We have revised the relevant statement and now refer to "increased the proximity between the transcription and replication machineries" (page 11).

To confirm equal DNA loading across samples in the dot blot assays we used methylene blue staining (see Figure EV4D). We normalized the S9.6 signal to the methylene blue staining and quantified two biological replicates, confirming that YAP5SA leads to a reduction in R-loop signal relative to uninduced cells (Figure EV4E). To independently test whether

YAP leads to R-loop formation, we performed a second method, MapR, which utilizes a catalytically inactive RNase H1 fused to micrococcal nuclease to map R-loops genome-wide. Treatment with the BRD4 inhibitor JQ1, which was done as a control, increased R-loops as expected, validating the assay. YAP5SA expression, either alone or in combination with CHK1 inhibition, did not lead to higher levels of R-loops detected by MapR. The new data are shown in Figure EV4F. We also added two new sections to the Methods part, describing first the purification of GST- Δ RH-MNase and secondly the MapR assay itself (page 18).

The MapR data have been submitted to GEO under the accession GSE296501: <https://www.ncbi.nlm.nih.gov/geo/query/acc.cgi?acc=GSE296501>. The reviewer token is: knizuwcefzuvvgn.

The reduced R-loop signal observed in the S9.6 based dot blot assay but lack of change in MapR could reflect differences in the sensitivity or specificity of the two methods. Importantly, both assays demonstrate that YAP expression does not increase R-loop levels, which supports the key conclusion of our study.

3. Following up on the above point, it is unclear why RNA:DNA hybrids seem reduced. Could upregulation of RNase H2 expression by YAP offer a simple mechanistic explanation? It should be quite straightforward to test this.

We analyzed the mRNA expression levels of RNASEH1 as well as the three subunits of RNASEH2 (A, B, and C) by RT-qPCR. Expression of these genes was not significantly altered in cells expressing YAP5SA. Given the lack of effect, we chose not to include these results in the manuscript. The data are provided here:

Figure for referees not shown.

4. In the description of figure 5K and 5L the authors claim that "inhibition of CDK4/6 activity and restoration of G1 length by palbociclib partially reduced the DNA-damage phenotype due to YAP expression". However they do not provide proof that G1 length

was actually restored by palbociclib. It seems to me that the palbociclib dosage is crucial in this experiment. If they blocked S-phase entry altogether in at least a subset of the cells with a high dose of palbociclib, they may also see lower DNA damage, simply because they arrest more cells in S-phase. I could not find what dosage they used in this experiment, so that should be mentioned in the figure (legend). But they should also show that average G1 duration is prolonged by palbociclib in this experimental setup, using flow cytometry or microscopy.

We thank the reviewer for this important comment. The experiment in Figure 5J was performed with 500 nM palbociclib, which increases the fraction of cells in G1, but does not completely block cell cycle entry, especially in YAP5SA expressing cells. We provide the cell cycle data based on EdU labeling and high content microscopy in Figure EV4G. We agree that since the experiment in Figure 5J was performed in asynchronous cell populations, the observed reduction in DNA damage could reflect changes in the proportion of cells in S-phase. To address this concern, we analyzed γ H2AX levels in S-phase by high content microscopy after restoring the G1 length of YAP5SA-expressing cells with a low concentration of 50 nM palbociclib. In Figure 2C we show that this concentration of palbociclib prolongs G1 without fully arresting S-phase entry (see Fig. 2C). We have updated the Results section and the Figure Legend to clarify the palbociclib dosages used in these experiments.

5. Following up on point 1: previous work also suggests that hyperactivation of YAP/TAZ causes replication stress and hypertranscription in neural progenitor cells (PMID:30523785). The authors should cite this paper, and place their work in the light of this existing literature, for example in the Discussion section.

We now cite this study in the Results and Discussion section on page 12 to acknowledge that YAP/TAZ hyperactivation induces replication stress and excessive transcriptional activity in neural progenitor cells.

6. Page 7 "This reduction in EdU... CHK1 activity [36-38]. The word "was" or "is" seems to be omitted after the word "which".

The mistake has been corrected.

7. Page 12 "The presence... replication fork to stall." The word "of" seems to be omitted after "presence".

The mistake has been corrected.

Referee #2

In figure 2C no statistical comparison is provided, and I believe there is a noticeable accumulation of cells in G2/M for the YAP5SA-overexpressing cells, that might be

significant. How do the authors explain this accumulation? Provide statistics and possible explanation in the text.

In the original submission, Figure 2C showed technical replicates from a representative experiment without statistical analysis. We have now included quantitative analysis across biological replicates and performed statistical comparisons of the cell cycle distributions. As shown in the updated Figure 2C, while YAP5SA-expressing cells treated with 100 nM palbociclib display a mild increase in the G2/M population, this difference is not statistically significant. Therefore, we do not consider the observed accumulation to be a consistent effect.

How does YAP5SA promote higher cyclin D-CDK4/6 activation? Is it at the protein levels of any of these components? Or does it have to do with an indirect effect through a different pathway regulating CDK4/6 activity? It would be a key point to add to Figure 2.

To address this question, we analyzed the expression levels of G1 cyclins (Cyclin D1 and Cyclin E) and their associated kinases (CDK4 and CDK6) by both RT-qPCR and western blotting. As shown in Figures EV1F and EV1G, YAP5SA expression leads to a moderate (~2-fold) upregulation of Cyclin D1 and CDK6 at both the mRNA and protein levels. Given that this induction is relatively modest, we cannot rule out the possibility that additional, indirect mechanisms contribute to the observed increase in CDK4/6 activity. We agree that elucidating the pathways underlying enhanced CDK4/6 activity in YAP-expressing cells would be of great interest, but we believe this question extends beyond the scope of the current study and should be addressed in future work.

Figure 2H, no error bars or statistical significance is shown in the figure. Please provide both in order to be able to conclude.

We thank the reviewer for this comment. We omitted error bars because the experiment shown in Figure 2H was performed twice. To better represent the variability between replicates, we have now updated the figure to include scatter plots displaying individual data points from both experiments.

In Figure 3A authors provide graphs of MCM7 intensity in early S phase. They establish tight thresholds to show MCM7+ cells but I believe some MCM7 levels should be present at late S-phase. I would suggest to re-analyze and complete this experiment. 1) Since G1 is the cell cycle stage where replication origin licensing finishes, I would show quantification of MCM7 levels in EdU-/low Hoechst (G1). Then provide QIBC plots as provided but setting the lower threshold of MCM7 to EdU- high Hoechst (G2) where eventually all MCM complexes should have been unloaded. Overall quantifications of G1, S and G2 could be provided; and then MCM7 levels in all S-phase cells could be represented as they have in Figure 3A. Further subdivision of the S-phase in early, mid, and late could be useful as well to understand MCM unloading dynamics.

We thank the reviewer for the suggestions to improve the analysis of MCM7 dynamics across the cell cycle. In response, we reanalyzed the data using a more accurate threshold for MCM7 positivity, based on its intensity in G2-phase cells (EdU⁻, high Hoechst), where MCM complexes are expected to be fully unloaded and staining should be minimal. Using this G2-based threshold, we replotted the MCM7 intensities in Figure 3A. A quantification of chromatin-bound MCM7 in early S-phase cells is provided in Figure 3B. Additionally, we include quantification of MCM7 levels in mid and late S-phase as well as in G2/M, as requested (Figure EV2C). Single cell data are presented as violin plots, allowing a more comprehensive view of MCM7 dynamics throughout the cell cycle independent from an arbitrary threshold.

In figure 3B, the % of positive cells would always depend on a threshold established by the authors, I would suggest plotting intensities of individual cells in a similar plot to the EdU intensity as in Figure 3C.

As requested, we plotted the MCM7 single cell intensities in violin plots, independently from an arbitrary threshold.

As already reported in literature (PMID: 29959228) origin firing and fork speed are interconnected. Probably the less availability of origins in YAP5SA-overexpressing cells makes faster forks. On the other hand, unrestrained origin firing in Chk1 inhibited cells, makes slower forks, counteracting the low origin availability in YAP5SA-overexpressing cells. This should be mentioned in the results or in the discussion.

We have now included this point and the reference in the Results and Discussion section on page 8. We now mention that in YAP5SA-overexpressing cells reduced origin availability may lead to compensatory acceleration of replication forks.

The proximity between PCNA and pRNA Pol II is being used as an indicative of TRC, but in fact, literature is full of examples where this does not correlate (PMID: 38581679, opposite trends in terms of TRC than that expected for the phenotype; PMID: 37468626, RNA Pol II association with replication outside the context of TRCs) as in the case of YAP5SA overexpressing cells. This is also in line with the absence of asymmetry, as observed in Figure 3H. Therefore, the fact that these cells suffer from TRCs is a statement that should be toned down to a "higher proximity between transcription and replication machineries".

We agree that that PCNA–RNA Pol II proximity should not be overinterpreted as direct evidence of transcription–replication conflicts (TRCs). As suggested, we have therefore revised the relevant statement and now refer to "increased the proximity between the transcription and replication machineries" (page 11).

In Supplementary Figure S5, the S9.6 could benefit of a control dot blot with the other half of the sample where the total amount of loaded DNA is shown for all conditions, as detected by anti-dsDNA (PMID: 36864174). Quantifications could then be performed using this control as reference.

To confirm equal DNA loading across samples in the dot blot assays, we used methylene blue staining (see Figure EV4D). The S9.6 signal was normalized to the methylene blue staining. Quantification from two biological replicates showed that YAP5SA expression leads to a reduction in R-loop signal relative to uninduced cells (Figure EV4E).

Since YAP5SA-overexpressing cells have DNA damage only in combination with Chk1 inhibition, does Chk1i trigger an increase in R-loops (dot blot) and TRC (PCNA:pRNA Pol II PLA) in YAP5SA-overexpressing cells?

To address this question, we performed MapR, which utilizes a catalytically inactive RNase H1 fused to micrococcal nuclease to map R-loops genome-wide. MapR showed that YAP5SA expression, either alone or in combination with CHK1 inhibition, does not increase R-loop formation (Figure EV4F).

In terms of TRCs, we have performed a PLA of PCNA and RNA PolII (pSer 2) and indeed observed a higher number of contacts when YAP5SA-expressing cells were treated with CHK1i compared to untreated cells (see Figure 5I).

Figure 5K is a WB of the whole cell population, thus upon treatment with Palbociclib cells accumulate in the G1, hence less S-phase cells and less signaling of DNA damage. This should be indicated in the text. The conclusion does not suffer since Figure 5L nicely shows a decrease in the S-phase.

We thank the reviewer for this important point. To clarify this point we now state on page 12: “Since these experiments were performed in asynchronous cell populations, the observed reduction in DNA damage could reflect a lower proportion of cells in S-phase rather than a direct effect on S-phase integrity”.

Could the sensitivity of YAP5SA-overexpressing cells to Chk1i and ATRi in terms of viability be rescued by CDK4/6 inhibition?

We appreciate the reviewer’s suggestion to test whether the sensitivity of YAP5SA-expressing cells to CHK1i and ATRi could be rescued by CDK4/6 inhibition. Initial test experiments showed that prolonged treatment with palbociclib, as required for the viability assays, is not feasible, as it leads to cell cycle arrest. This arrest would likely confound the interpretation of the results, as it masks the effect of CHK1i or ATRi treatment. Nonetheless, we agree that this is an interesting question, and we plan to explore this in future studies.

Previous literature on the role of YAP in DNA replication in different organisms is not mentioned e.g. PMID: 35838349, which proposes a role for YAP together with RIF1 to

regulate DNA replication program in Xenopus. I believe this should be included in the text in the Introduction section. Moreover, how much of the effects observed in YAP5SA has to do with its interaction with RIF. Depletion of RIF1 might be tested upon expression of YAP5SA in key experiments to complement the story and provide further mechanistic detail.

We thank the reviewer for highlighting this study, which we now cite and discuss in the Introduction to acknowledge previous work linking YAP and DNA replication regulation via RIF1 in Xenopus. To explore whether a similar mechanism may be at play in our system, we tested whether RIF1 depletion affects DNA damage levels in YAP5SA-expressing MCF10A cells. siRNA-mediated knockdown of RIF1 did not lead to increased DNA damage in YAP5SA-expressing cells, as assessed by γ H2AX immunostaining. As this is a negative result and does not support a functional interaction between RIF1 and YAP5SA under our experimental conditions, we suggest not to include these data in the revised manuscript. The data are included for the reviewers here:

Figure for referees not shown.

YAP is defined as "Yes-associated protein" in page 4 while it has been introduced before either abstract or first word of the introduction

We now define YAP in the first sentence of the introduction (page 2)

First title in Results, I believe authors meant "Oncogenic" instead of "Oncogenes".

The mistake has been corrected, and "oncogenes" has been changed to "oncogenic".

Cytoplasmic endogenous YAP is barely visible in Fig 1A, please provide a different image where this is properly shown.

As endogenous YAP levels are low in MCF10A cells, it is challenging to detect without altering the image settings, which we wanted to avoid. We now provide a different image in which cytoplasmic endogenous YAP is a bit more clearly visible. Additionally, we have included the raw image data with the revised submission.

I would suggest changing "YAP-expressing/overexpressing cells" by "YAP5SA-overexpressing cells " to avoid confusion. Same for the labeling of the figures.

We have changed YAP-expressing/ overexpressing to YAP5SA-expressing throughout the text and Figures as requested.

Figure Legend 1. 10 nM EdU (I believe this is a mistake and it should be 10 μ M) while in methods it states "treated with 10 mM EdU". I recommend stating "labeled instead of treated" and write the final concentration (10 μ M?) not the stock solution (10 mM?)

This was indeed a mistake and a final concentration of EdU used was 10 μ M. This has now been corrected in both the Figure Legend and the Methods section. We have also replaced "treated" with "labeled" to more accurately describe the procedure.

Representative images of both reporters might be shown in Figure 2B, at least for the first time it is used in the manuscript. This should be done as well for Fig. 3A, 4B and 5J.

We show representative images of the reporters and IF data in the following Figures:

EV1A: DHB-mVenus and CDK4KTR-mcherry

EV2B: MCM7

EV3A: γ H2AX

EV4B: PIP-FUCCI

EV4C: PLA

Page 6: when mentioning "(Figure 2K,L)" add spacing before L and full stop "." right after parenthesis.

The mistake has been corrected.

Rephrase 3rd title of Results.

We have changed the title to "YAP5SA leads to underlicensing and accelerated DNA-replication".

The authors show a dependency on CHK1 kinase activity to restrain EdU incorporation. Do YAP5SA-overexpressing cells have a higher activation Chk1 kinase?

To assess CHK1 activation, we performed western blotting using an antibody specific for CHK1 phosphorylated at serine 296 (pS296), a marker of CHK1 kinase activity. We observed a slight increase in CHK1 pS296 levels in YAP5SA-expressing cells, which may contribute to the reduced EdU incorporation observed under these conditions. The corresponding western blot is now included in EV2D, and the results are described in the revised manuscript on page 8.

A bit more detail in the text on how DNA fibers are labelled and produced (before mentioning Figure 3F) could be helpful for non-experts.

We provide more details on the DNA fiber assays on page 8.

Typo in Figure 4D "y axis".

The typo has been corrected.

In figure 4G, I would not state that damaged cells are "stalled in mid replication" since EdU is being incorporated albeit at lower levels. This phenotype is more indicative of a severe replication defect than a complete stall.

We agree and have we changed the text on page 10 to "Consistent with this notion, prolonged CHK1 inhibition (4 hours and 8 hours) led to severe replication defects with low levels of EdU incorporation but with high levels of the DNA-damage markers γ H2AX and pKAP1 (Fig. 4G)."

Third lane of "Oncogenic YAP sensitizes cells to CHK1 and ATR inhibition" section, there is Chk1 "inhibitor" missing.

The mistake has been corrected.

The % of gammaH2AX positive cells in S-phase in Figure 5I could be substituted for a plot of gamma intensity in these cells avoiding establishing arbitrary thresholds

As suggested, we have replaced the plot depicting % gammaH2AX positive cells (Figure 5I) by a violin plot to avoid setting a threshold (new Figure 5H).

Referee #3

CDK4/6 activity calculation: Given that KTR-based sensors were designed using the CDK2 sensor, a correction factor needs to be applied to the CDK4/6 sensor. For MCF-10A cells, the correction factor has been previously calculated. Corrected CDK4/6 activity = CDK4/6 reporter activity - 0.35*CDK2 reporter activity.

We thank the reviewer for pointing out that KTR-based CDK4/6 activity sensors require correction due to partial responsiveness to CDK2. In our analysis, we had already applied the previously established correction factor for MCF10A cells: Corrected CDK4/6 activity = CDK4/6 reporter activity - 0.35 × CDK2 reporter activity. We have now updated the Materials and Methods section to explicitly describe how CDK4/6 activity was measured and calculated (page 15/16).

Figure 2B-CDK2 Activity and YAP Overexpression: Since CDK2 activity facilitates the G1/S transition, it is important to evaluate its dynamics throughout the cell cycle. If Figure 2B measurements focus only on G1-phase cells, the potential for no observed differences in CDK2 activity despite YAP overexpression remains. A more detailed analysis of CDK2 activity across different phases, particularly in the context of YAP overexpression, would enhance the conclusions.

To address this concern, we analyzed CDK2 activity across additional cell cycle phases. Specifically, we quantified CDK2 activity in S-phase and G2/M cells and included this analysis in Figure EV1B. Consistent with our G1-phase data, we did not observe strong differences in CDK2 activity during S or G2/M phases upon YAP5SA expression. These findings suggest that YAP does not broadly alter CDK2 activity throughout the cell cycle.

Figure 2E - Rb Phosphorylation Interpretation: Rb phosphorylation can be maintained in S-phase cells due to the restriction point. Therefore, S-phase cells should retain Rb phosphorylation regardless of CDK4/6i treatment. Do they indeed maintain Rb phosphorylation, but the distinction between phosphorylated and non-phosphorylated populations becomes less clear after CDK4/6i treatment? Additionally, does YAP overexpression further enhance this separation in the presence of CDK4/6i? Clarifying these points would help in interpreting the results more effectively.

We agree with the reviewer that Rb phosphorylation is expected to persist in S phase due to the passage through the restriction point. Consistent with this, we observed that palbociclib treatment had a much weaker effect on the pRB/RB ratio in S-phase cells compared to G1 (Figure EV1D), indicating that most S-phase cells maintain Rb phosphorylation despite treatment. Interestingly, YAP5SA expression led to a modest reduction in the pRB/RB ratio in S-phase cells, even in the presence of palbociclib. Further investigation will be required to determine the mechanism underlying this reduction.

Figure 2F-L - the potential impact of YAP overexpression on CDK2 activity: These results also be explained by the potential of CDK2 activity promotion by YAP overexpression.

We did not observe a significant change in CDK2 activity upon YAP overexpression (Figure 2B and Figure EV1B). Therefore, while we cannot entirely exclude indirect effects, a direct promotion of CDK2 activity by YAP overexpression appears unlikely in our system.

Figure 5G - Potential Enhancement of E2F and CDK2 Activity: YAP overexpression may increase global mRNA transcription and E2F activity, thereby promoting CDK2 activation and facilitating the G1/S transition.

We agree that YAP may drive global mRNA transcription and E2F activity, but since we did not observe effects on CDK2 activity, the accelerated G1/S transition is likely driven by increased CDK4/6 activity rather than CDK2.

Figure 4D y-axis Label Correction: "Cytoplanic" should be corrected to "cytoplasmic."

The mistake has been corrected.

Scatter plots (e.g. Figure 4B): The number of plotted cells should be consistent across different conditions to allow for direct comparisons. Additionally, the total number of cells analyzed should be indicated in the figure legend.

In the updated scatter plots (e.g., Figure 4B), we now display the same number of randomly selected cells per condition to allow for direct visual comparison across samples. Additionally, the total number of cells analyzed per condition is now indicated in the corresponding Figure Legend.

Dear Dr. Gaubatz

Thank you for the submission of your revised manuscript. We have now received the enclosed reports from the referees. Referee 1 still has a few more minor suggestions that I would like you to incorporate before we can proceed with the official acceptance of your manuscript.

A few editorial requests will also need to be addressed:

- Please add up to 5 keywords to the ms file.
- The Disclosure and Competing Interest Statement needs to be placed after the Acknowledgments
- The author credits need to be removed from the ms file. All credits need to be entered during online ms submission.
- The REFERENCES need to be alphabetical, not numerical; et al needs to be used after 10 author names. Please use the EMBO reports reference style.
- A CALLOUT for Fig 5KL is missing in the ms text, please add.
- The legend for Table EV1 needs to be removed from the ms file.
- Please remove the instructions and example table from the Reagents & Tools table file.
- Figures 5H, EV3 C, EV4 E state n=2 but statistics are calculated. This is not possible. Please either repeat the experiment at least one more time or remove the statistics. For n=2 please show all data points along with their mean.

Figure Legends - Comments

- Please note that the exact p values are not provided in the legends of figures 1B, 2B, E; 3B, C, D, E, G; 4C, 5B, G-I, K; EV1 B, D; EV2 C, EV3 E. Please provide exact p-values as reasonable.
- Please indicate the statistical test used for data analysis in the legends of figures 2B, EV1 B, EV3 E.
- Please note that the error bars are not defined in the legends of figures 1D, 2C, G; 4C-F; 5A-F; EV3 E, EV4 A.

Referee #1:

I have read the point-by-point rebuttal and the revised manuscript. Gertzmann and colleagues have addressed my points carefully and they clearly improved the paper. I only have a couple of remaining minor comments / requests that can be easily addressed:

RE point 1. The approach to use PIP-FUCCI to account for potential cell cycle bias -instead of the IdU that I suggested- is in principle fine with me, and I am glad to see that the increase in YAP5SA expressing cells is still present. However, to my knowledge the PIP-FUCCI system is not able to distinguish between S- and G2- phase cells. Hence they exclude G1 cells and they quantify S+G2 cells in 5G (instead of only looking at S-phase cells). This is fine and sufficiently addresses my main concern, i.e. that global transcription is lower during G1. However I recommend that they mention this small limitation in the text on page 11.

RE point 3. I thank the authors for exploring my suggestion. I agree that it is not necessary to include this in the manuscript.

RE point 4. This clarification is helpful and strengthens the manuscript. However, I see that EV4G is mentioned in the text and figure legend, but I cannot find a panel EV4G in the figure. I suppose there has been a mix-up with versions of the figure plates during the revision process and resubmission? Can the authors please provide these cell cycle data?

Referee #2:

The authors have addressed properly my concerns and I support its publication in EMBO reports.

Referee #3:

I have no further concerns.

EMBOR-2025-61320V2

We have incorporated the suggestions from Reviewer 1 and made the requested editorial changes as outlined below:

Referee 1:

RE point 1. The approach to use PIP-FUCCI to account for potential cell cycle bias -instead of the IdU that I suggested- is in principle fine with me, and I am glad to see that the increase in YAP5SA expressing cells is still present. However, to my knowledge the PIP-FUCCI system is not able to distinguish between S- and G2- phase cells. Hence they exclude G1 cells and they quantify S+G2 cells in 5G (instead of only looking at S-phase cells). This is fine and sufficiently addresses my main concern, i.e. that global transcription is lower during G1. However I recommend that they mention this small limitation in the text on page 11.

It is correct that it is difficult to distinguish between S-phase and early G2 using PIP-FUCCI, as the green fluorescence gradually increases after S-phase. We have updated the text on page 12 accordingly to state that we gated cells for S-phase and early G2.

RE point 4. This clarification is helpful and strengthens the manuscript. However, I see that EV4G is mentioned in the text and figure legend, but I cannot find a panel EV4G in the figure. I suppose there has been a mix-up with versions of the figure plates during the revision process and resubmission? Can the authors please provide these cell cycle data?

We apologize for the omission of panel EV4G in Figure EV4 during the upload of the re-submission. The correct figure, including panel EV4G, has now been uploaded.

Additionally, we have made the requested editorial changes:

- We have added keywords to the ms file.
- The Disclosure and Competing Interest Statement has been placed after the Acknowledgments.
- The author contributions were removed from the ms file.
- We now use the *EMBO Reports* reference style.
- Figure 5K is mentioned in the text.
- The legend for Table EV1 was removed from the ms file.
- The example table and instructions were removed from the Reagents & Tools table file.
- In Figures 5H, EV3 C, EV4 E the statistics was removed.

Figures and Figure Legends:

- We give the exact p-values for values >0.0001 . For very small p-values, GraphPad Prism reports values as <0.0001 , because the actual p-value cannot be precisely estimated. Therefore, for values below 0.0001, p-values are indicated as " <0.0001 ". We have added a note regarding the p-values to the Methods section (see page 20).
- Statistical test used for data analysis are now provided for legends of figures 2B, EV1B, EV3E.
- Error bars are defined in the legends of figures 1D, 2C, G; 4C-F; 5A-F; EV3 E, EV4 A.

Dr. Stefan Gaubatz
University of Wuerzburg
Biocenter
Biochemistry and Cell Biology
Am Hubland
Würzburg, Bavaria 97074
Germany

Dear Dr. Gaubatz,

I am very pleased to accept your manuscript for publication in the next available issue of EMBO reports. Thank you for your contribution to our journal.
